# A new perspective on population genetics: Deciphering the relationship between genetic variants and disease prevalence in Psoriasis

Yuanjing Zhang[1,2], Weiran Li[2,3,4,5], Wanrong Wang[6], Kejia Wu[6], Feiran Zhou[6], Xiaodong Zheng [2,3,4,5]*

**1** Department of Dermatology, The First Affiliated Hospital of USTC, Division of Life Sciences and Medicine, University of Science and Technology of China, Hefei, Anhui, China, **2** Department of Dermatology, the First Affiliated Hospital of Anhui Medical University, Institute of Dermatology, Anhui Medical University, Hefei, China, **3** Key Laboratory of Dermatology (Anhui Medical University), Ministry of Education, Hefei, China, **4** Inflammation and Immune Mediated Diseases Laboratory of Anhui Province, Hefei, China, **5** Anhui Provincial Institute of Translational Medicine, Hefei, China, **6** First Clinical Medical College, Anhui Medical University, Hefei, Anhui Province, China

* zhengxiaodong@ahmu.edu.cn

## Abstract

In the quest to identify the genetic underpinnings of complex diseases, we developed a novel approach called Causal Genotype Combination Patterns (CGCP) to uncover characteristic genetic signatures of common diseases. In this study, we applied the CGCP method to a whole-exome sequencing dataset of 781 psoriasis cases and 676 healthy controls from the Chinese Han population. Our analysis revealed 620 genotype combinations specific to psoriasis, covering 4.7% to 10% of all cases, with each genotype having a frequency of at least 1%. These genotypes converged into 134 genes, including 41 previously reported to be associated with psoriasis. By leveraging public data from the 1000 Genomes Project Phase III and literature reviews on psoriasis prevalence in various ethnic populations, we established a strong positive correlation and linear regression model ($y = 61.72x + 0.48$, 95% CI [21.60, 101.84]) between the average frequency of these psoriasis-specific genotype combinations and disease prevalence across populations. This finding may explain the varying prevalence of psoriasis in different populations. Our strategy offers a new perspective on understanding the characteristics of population genetics in common diseases.

## Introduction

Psoriasis (MIM: #177900) is a chronic, immune-mediated inflammatory skin disease with a significant genetic predisposition. Its prevalence varies widely across different regions, with rates of 1.5-2.8% in the United Kingdom, 3.7% in the United States, and 1.9–3.5% in Eastern Africa compared to lower rates of 0.03–0.9% in Western Africa and 0.05-0.47% in Asia [1,2]. Although the prevalence of psoriasis varies globally, it

**Data availability statement:** All relevant data are within the paper, its Supporting Information files. The CGCP code is available on GitHub at github.com/XiaoDongZheng1234/CGCPs.

**Funding:** This study was supported by the Basic and Clinical Cooperative Research Promotion Program of Anhui Medical University (2023xkjT047, awarded to XZ), Key project of Anhui Provincial Department of Education (KJ2021A0267, 2022AH051155, awarded to XZ), Key Laboratory of Dermatology, Anhui Medical University, Ministry of Education, China (Number. AYPYS2021-2, Number. AYPYS-2023-1, awarded to YZ), Anhui Province Key Research and Development Program (2023s07020021, awarded to XZ), Wuhu City Industrialization Project for Patented Invention and Technological Achievements (202419, awarded to XZ), and the Anhui 'Tongxin' Scientific and Technological Innovation Project (Grant No. 202523b11020015, awarded to XZ). The funders had no role in study design, data collection and analysis, decision to publish, or preparation of the manuscript.

**Competing interests:** This study was supported by the Basic and Clinical Cooperative Research Promotion Program of Anhui Medical University (2023xkjT047), Key project of Anhui Provincial Department of Education (2022AH051155). Key Laboratory of Dermatology, Anhui Medical University, Ministry of Education, China (Number.AYPYS2021-2).

generally shows an apparent upward trend. Specifically, from 1984 to 2009, the prevalence in China increased from 0.17% to 0.47% [3,4].

In the United States, the prevalence of psoriasis increased from 1.62% in 2004 to 3.01% six years later [5,6]. The most informative study on prevalence trends is a 30-year follow-up of a population-based cohort in Norway. During the period from 1979 to 2008, the self-reported lifetime prevalence of psoriasis increased from 4.8% to 11.4% [7].

Genetically, the prevalence of psoriasis varies among different ethnicities, partly due to the heterogeneity of genetic architecture. The gradually increasing prevalence can be attributed to the cumulative effect of risk alleles [8,9].

To date, over 60 susceptibility regions associated with psoriasis have been identified through genome-wide association studies (http://www.ebi.ac.uk/gwas/), including HLA-Cw06:02, which has been observed in various ancestral groups. While GWAS can aid in understanding the genetic architecture of complex diseases, these susceptibility loci collectively only account for a limited fraction of psoriasis heritability. This "missing heritability problem" arises from variants with effects too small to achieve genome-wide significance [10–13]. The aggregation of these small effects is significant in complex diseases, and there may be critical combinations of risk alleles involved in the genetic basis of the disease.

In our previous study, we devised a novel method to identify causal genotype combination patterns (CGCPs) for common diseases, based on two fundamental genetic principles. One is that complex diseases arise from multiple genes in a specific genotype pattern, and the other is that the sum of the frequencies of all causal genotype patterns in a population will approximately equal the disease's prevalence [14]. Here, we propose a novel strategy based on the CGCP method to identify the specific genetic signatures of psoriasis in combination format. We utilized a whole-exome sequencing dataset of psoriasis generated from our previous study and incorporated data from the 100 Genomes Project phase 3 as a public data source [15]. Through this study, our aim is to uncover the genetic signature of psoriasis and investigate the genetic mechanisms contributing to its diverse manifestations and increased prevalence across different populations.

## Materials and methods

### Study population and the data source

We conducted CGCP analysis using whole-exome sequencing data from 781 Chinese psoriasis cases and 676 controls which was obtained by our previously study published in 2013 [15]. The medical ethics committee of the First Affiliated Hospital of Anhui Medical University approved the protocol (approval number: 2023258), and all patients provided written informed consent.

Genomic DNA from individuals was subjected to hybridization with the NimbleGen2.1 M-probe sequence capture array, and a total of 1457 exomes were subsequently sequenced. Each individual had an average coverage of 34-fold, and sequences were generated as 90-bp reads (Roche Inc. Denmark). Additionally, we utilized the 1000 Genomes Project phase III dataset which encompassed different

ethnicities including JPT (Japanese in Tokyo, Japan), CHB (Han Chinese in Beijing, China), YRI (Yoruba in Ibadan, Nigeria), ASW (Americans of African Ancestry in Southwest USA), GBR (British in England and Scotland), IBS (Iberian Population in Spain), ITU (Indian Telugu from the UK), CEU (Utah Residents with Northern and Western European Ancestry) and TSI (Toscani in Italia). The purpose was to investigate whether a correlation existed between disease prevalence and the frequency of disease-specific genotypes.

## Genotype combination analysis

Genotype combination analysis was conducted using the CGCP method, which we previously developed based on an exhaustive algorithm [14]. In this study, we defined the prevalence threshold as 0.47% based on a recent epidemiological study of the Chinese Han population, which had a large sample size. In order to manage the size of the output file, our program only included combinations that were observed at least 10 times in cases but not once in controls.

The total number of genotype combination patterns, denoted as N, based on the eligible single nucleotide polymorphisms (SNPs), was determined as follows:

$$N = 3r\frac{n!}{(n-r)!\,r!}$$

(1)

where n is the total number of eligible SNPs and r is the number of selected SNPs assigned to produce genotype combinations, which in this study is set as 3. The real causal combinations of psoriasis may be built up by four, five, six, or more genotype combinations. In this study, we started with 3 genotype combinations to reduce the computational cost.

The estimator of the population genotype frequency, denoted as $\hat{F}_i$, is defined as:

$$\hat{F}_i = (\alpha i + \lambda \beta i)/(1 + \lambda);$$

(2)

Where the subscript i represents the index of the i-th genotype. The parameter $\lambda$ is equal to $(1 - P)/P$, where P represents the prevalence rate of the population. The terms $\alpha_i$ and $\beta_i$ denote the genotype frequencies in the case and control, respectively.

The prevalence rate (P) was further used as a boundary condition to filter the combination, for a selection of r SNPs. The joint estimator is calculated as follows, where the product is taken over the index i:

$$0.01\,P < \prod_{i=1}^{r} \hat{F}_i < P;$$

(3)

The CGCP method is a script written in Perl version 5.01 (Inc. 51 Franklin St, Fifth Floor, Boston, MA 02110−1301 USA), and it was described in detail in our previous study. Related programs have also been published on the GitHub website (https://github.com/XiaoDongZheng1234/CGCPs).

## The background noise of CGCPs

To evaluate the background noise in CGCP calculations, we performed analyses using samples and SNPs unrelated to psoriasis. First, the 676 healthy controls were randomly divided into two groups, with one group designated as "patients." CGCP analysis was then conducted using variants selected from psoriasis-associated loci. Additionally, in an independent dataset comprising 781 cases and 676 controls, CGCP analysis was performed using SNPs located outside the psoriasis-associated regions, thereby capturing background noise originating from SNPs unrelated to psoriasis. The threshold for background noise was determined by comparing the maximum number of "specific" combinations derived from these two unrelated datasets (Fig 1).

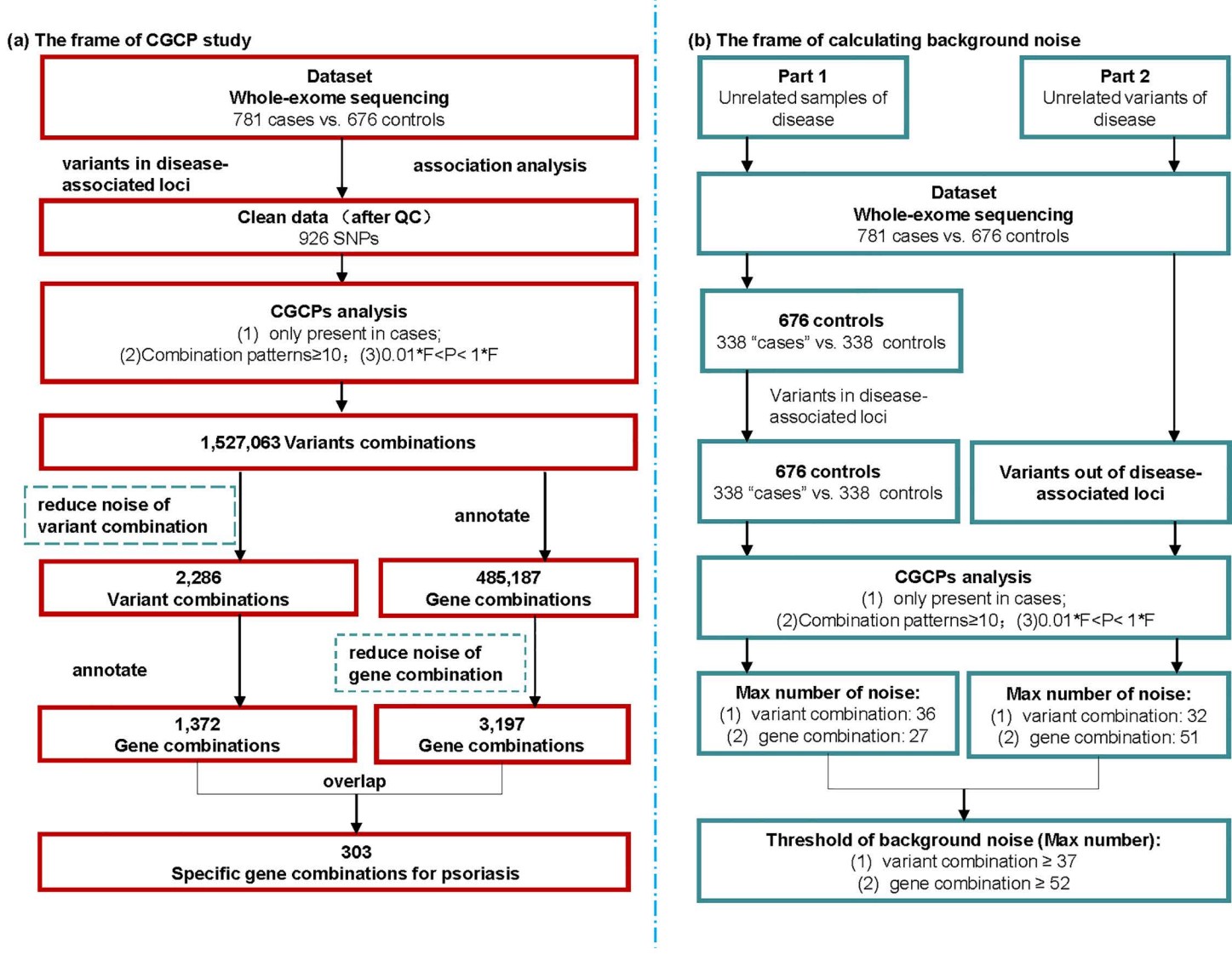

**Fig 1. Flow diagram of the CGCP study.** Overview of the CGCP study framework, illustrating the analysis pipeline for eligible SNPs from exon sequencing data. (A) Depicts the CGCP study framework; (B) illustrates the calculation of background noise, with F representing genotype frequency and P representing disease prevalence.

### Statistical analysis for exploring the relationship between the frequency and prevalence

To assess the correlation between the frequency of psoriasis-specific genotype combinations and disease prevalence in different ethnicities, we used the Pearson correlation coefficient. This statistical analysis was performed using SPSS software (v.23.; IBM, Armonk, New York, USA). In addition, we conducted computer simulations in three models to further investigate this relationship. The details of these simulations and the results were described in the supplementary file called.Supplementary _computer program simulation.

## Stability test with computer simulation

The real causal combinations of psoriasis might include more than three combinations. To make the simulation test have good operability, we assumed that the number of causal combinations of psoriasis was three and consisted of 4, 5 and 6 genotypes. The combined frequency can be defined as $f_4$ (the frequency of each genotype was defined as $a_1$, $a_2$, $a_3$, $a_4$), $f_5$ (the frequency of each genotype was defined as $b_1$, $b_2$, $b_3$, $b_4$, $b_5$) and $f_6$ (the frequency of each genotype was defined as $c_1$, $c_2$, $c_3$, $c_4$, $c_5$, $c_6$). In this case, the prevalence of disease (P) satisfies the following equation:

$P = f_4 + f_5 + f_6 = a_1a_2a_3a_4 + b_1b_2b_3b_4b_5 + c_1c_2c_3c_4c_5c_6$. In each test, fifteen numbers are randomly generated in the range (0, 1), representing the genotype frequencies of 15 variants ($a_1$, $a_2$, $a_3$, $a_4$, $b_1$, $b_2$, $b_3$, $b_4$, $b_5$, $c_1$, $c_2$, $c_3$, $c_4$, $c_5$, $c_6$) in the first population, and the corresponding P value will be calculated programmatically. Simultaneously, these 15 random numbers are combined in groups of three to compute the frequencies of three-locus genotype combinations (denoted $f_{-4}$, $f_{-5}$, $f_{-6}$ for groups $a_1a_2a_3a_4$, $b_1b_2b_3b_4b_5$, $c_1c_2c_3c_4c_5c_6$, respectively). The average frequency, $\overline{X}$, is calculated across 34 three-genotype combinations, including $a_1a_2a_3$, $a_1a_2a_4$, $a_1a_3a_4$, $a_2a_3a_4$; $b_1b_2b_3$, $b_1b_2b_4$, …; $c_1c_2c_3$, $c_1c_2c_4$, …, following the same pattern.. When we run the test in N groups once (the number of populations was defined as N), N sets of P and corresponding $\overline{X}$ were output together. We used linear least-square regression to detect whether there is a significant correlation between $\overline{X}$ and P, and this was repeated 1000 times to test for robustness. We further restricted the range of 15 random numbers in four different ways to comprehensively simulate the relationship between P and $\overline{X}$. The details of all test models were shown on the supplementary file called Supplementary _computer program simulation.

## Results

### CGCP survey with whole-exome sequencing data in 781 cases and 676 controls

Following association analysis, a total of 948 candidate genetic markers (0.9%) were included in the CGCP analysis based on quality control standards (S1 Table). Through CGCP analysis, it was discovered that there were 1,527,063 genotype combinations exclusively present in psoriasis patients, with 70,562 combinations observed in more than 20 patients (S2 Table). These 1,527,063 genotype combinations were then combined into a total of 485,187 combinations of variants of genes.

The characteristics of the individuals in the whole-exome sequencing data are presented in Table 1. A total of 650,041 single nucleotide polymorphisms (SNPs) were identified in the exome sequencing data, with 101,657 SNPs (15.64%) falling within psoriasis-associated regions from the GWAS catalog (S3-S4 Tables). After conducting association analysis, a subset of 948 candidate genetic markers (0.9% of SNPs) that met quality control criteria were included in the CGCP analysis (S1 Table). The CGCP analysis revealed that there were a total of 1,527,063 genotype combinations specific to patients with psoriasis, out of which 70,562 combinations were observed in more than 20 patients (S2 Table). All these genotype combinations were then merged into a total of 485,187 combinations of variants of genes.

**Table 1. The characteristics of individuals in whole exome sequencing data.**

| Characteristic | Case (n=781) | Control (n=676) |
|---|---|---|
| Gender | | |
| Male, n (%) | 458 (58.64%) | 407 (60.21%) |
| Female, n (%) | 323 (41.36%) | 269 (39.79%) |
| Age | | |
| mean (SD), y | 30.92(12.47) | 34.12 (12.92) |
| Age at disease onset | | |
| mean (SD), y | 22.45(9.39) | / |

SD, standard deviation. Mean, the mean value of Age/Age at disease onset. Y, year.

## Investigation of background noise

In the first scenario, CGCP analysis was performed on 338 "cases" and 338 controls, involving 482 SNPs, that showed significant differences between cases and controls after quality control (S5 Table). A total of 9,802 genotype combinations considered specific to psoriasis were identified, with the maximum number of specific combinations being 16 (4.7%) (S6 Table). After annotating the SNPs in these combinations to their corresponding genes, 4,640 identical combinations of variants of genes were obtained, among which the maximum number of specific combinations of variants of genes was 12 (3.6%) (S7 Table).

In the second scenario, association analysis was conducted using 548,357 SNPs located outside psoriasis-associated regions, based on whole-exome sequencing data. After quality control and pruning, 570 SNPs were included in the CGCP analysis (S8 Table). A total of 63,657 genotype combinations were identified as specific to psoriasis, with the maximum observed number of specific combinations being 32 (4.1%) (S9 Table). After annotating the SNPs in these psoriasis-specific combinations to genes, 62,971 combinations of variants of genes were obtained, and the maximum number of specific combinations of variants of genes was 52 (6.5%) (S10 Table).

Based on the scale-up principle, thresholds for investigating background noise in psoriasis-specific genotype combinations and combinations of variants of genes were established. Specifically, the threshold for genotype combinations was set at 37, calculated as $[(781/338) \times 16]$, and that for combinations of variants of genes was set at 52.

## Psoriasis-specific genotype and combinations of variants of genes were identified by the CGCP program

After filtering out background noise, we identified a total of 2,285 psoriasis-specific genotype combinations and 3,197 psoriasis-specific combinations of variants of genes. Through annotating these genotype combinations into their corresponding genes, we identified 1,373 combinations of variants of genes that were specific to psoriasis (S11 Table). Among these combinations of variants of genes, we found 303 unique combinations consisting of 134 different genes. There was an overlap between the combinations of variants of genes obtained from different analyses (3,197 and 1,373), resulting in a total of 620 genotype combinations (S12 Table). The framework of our CGCP study is illustrated in Fig 1.

## Positive correlation between the frequency of psoriasis-specific genotype combinations and the prevalence of disease in different ethnicities

Using the CGCP analysis, we identified 620 genotype combinations that are specific to psoriasis and consist of 357 single SNPs. Out of these SNPs, 349 were present in the 1000 Genomes Project phase III dataset, which included a total of 600 psoriasis-specific genotype combinations. We calculated the frequency distribution of these specific genotype combinations in different ethnic populations within the dataset, including JPT, CHB, YRI, ASW, GBR, IBS, ITU, CEU and TSI (S13 Table). The frequency distribution for these genotype combinations showed a positive skew (S1 Fig).

The prevalence of psoriasis in different populations was obtained through a thorough literature review (Table 2). Correlation analysis revealed a significant positive correlation between the average frequency of psoriasis-specific genotype combinations and the prevalence of the disease in various populations. The median value exhibited a stronger correlation, with a Pearson correlation coefficient of 0.809 and a p-value of 0.008, compared to the mean value, which had a Pearson correlation coefficient of 0.776 and a p-value of 0.014 (Fig 2).

We further conducted a linear regression analysis to explore the relationship between the prevalence of psoriasis and the median frequency of psoriasis-specific genotype combinations. The obtained equation for linear regression was $y = 61.72x + 0.48$, 95% CI [21.60, 101.84], where y represents the prevalence of psoriasis and x represents the median frequency of psoriasis-specific genotype combinations. Moreover, we investigated the correlation between the prevalence of psoriasis in different ethnicities and the average value of genotype combinations derived from two background noise datasets (S14 Table). However, no significant correlation was observed in these comparisons (P > 0.05) (S2 Fig).

 

**Table 2. The prevalence of psoriasis in different populations.**

| Abbreviation of Ethnic[a] | Full name | Area/Hospital/Ethnic[b] | Population studied | Subjects | Patients | Preva-lence (%) | Period/point | PMID |
|---|---|---|---|---|---|---|---|---|
| JPT | Japanese in Tokyo, Japan | Japan | population-based | 128,000,000 | 429,679 | 0.34 | 2010 (Oct) | 25588781 |
| CHB | Han Chinese in Beijing, China | Taiyuan, Langfang in Hebei, Hailar, Zibo, Jiaozuo in Henan province, and Xichang city, China | population-based | 17,345 | 102 | 0.47 | 2007-2008 | 22910173 |
| YRI | Yoruba in Ibadan, Nigeria | University College Hospital, Ibadan, Nigeria | clinic-based | 1,091 | 10 | 0.9 | 1994-1998 | 14693018 |
| ASW | Americans of African Ancestry in Southwest USA | African Americans | population-based | 2,443 | 27 | 1.1 | 2001 (Nov to Dec) | 15627076 |
| GBR | British in England and Scotland | United Kingdom | population-based | 7,533,475 | 114,521 | 1.5 | 1987-2002 | 16365254 |
| IBS | Iberian Population in Spain | Northern Spain | population-based | 2,138 | 32 | 1.5 | 1998 | 11451315 |
| ITU | Indian Telugu from the UK | Tertiary health care center from North India | clinic-based | 52,477 | 1,120 | 2.3 | 1989-1993 | 9164063 |
| CEU | Utah Residents with Northern and Western European Ancestry | United States white population | population-based | 21,921 | 541 | 2.5 | 2001 | 15083780 |
| TSI | Toscani in Italia | Latium, Marches, Tuscany, Umbria, centre city of Italy | population-based | 919 | 31 | 3.4 | 2006 | 18269600 |

[a], Ethnic reported in 1000 genome project; [b], area/hospital/ethnic in epidemiological survey;

### The permutation test revealed a positive correlation between the average frequency of causal genotype combinations and prevalence of disease

We conducted four different permutation test models to investigate the relationship between the average combination frequency and prevalence. Overall, regardless of the model used, as the number of simulated populations increased, the linear relationship became more stable. To better reflect real-world scenarios and minimize Type I and Type II errors, we designed three additional models: "Constrained model," "Constrained lost model," and "Constrained lost and mixed model." In each model, we adjusted parameters related to mixed and lost proportions to create different subgroups and simulated a total of 36 scenarios. In all simulations, we observed that the $R^2$ value in the random model was generally lower than in other models. Additionally, with an increase in simulated population size in the random model, the $R^2$ value decreased significantly, indicating that the linear relationship generated by this random model was highly unstable. Conversely, in all three models that closely resembled real-world situations, both $R^2$ values and P values were more stable (Table 3).

In the last three models, we observed that all the models had positive slopes, indicating a positive linear correlation between P (prevalence) and $\overline{X}$ (average frequency). The results of the permutation test demonstrated that, under specific conditions, there is a stable linear correlation between the average frequency of multiple combinations containing at least three genotypes and the prevalence of common diseases when there is a sufficiently large population size (Fig 3). More detailed results for each model can be found in S15–S23 Table.

### Characteristics of variants in psoriasis-specific genotype combinations

The permutation test and mathematical derivation (Supplementary file_computer program simulation (S1 File) and Supplementary file_mathematical derivation (S2 File)) demonstrated that the genotype combinations identified by the CGCP

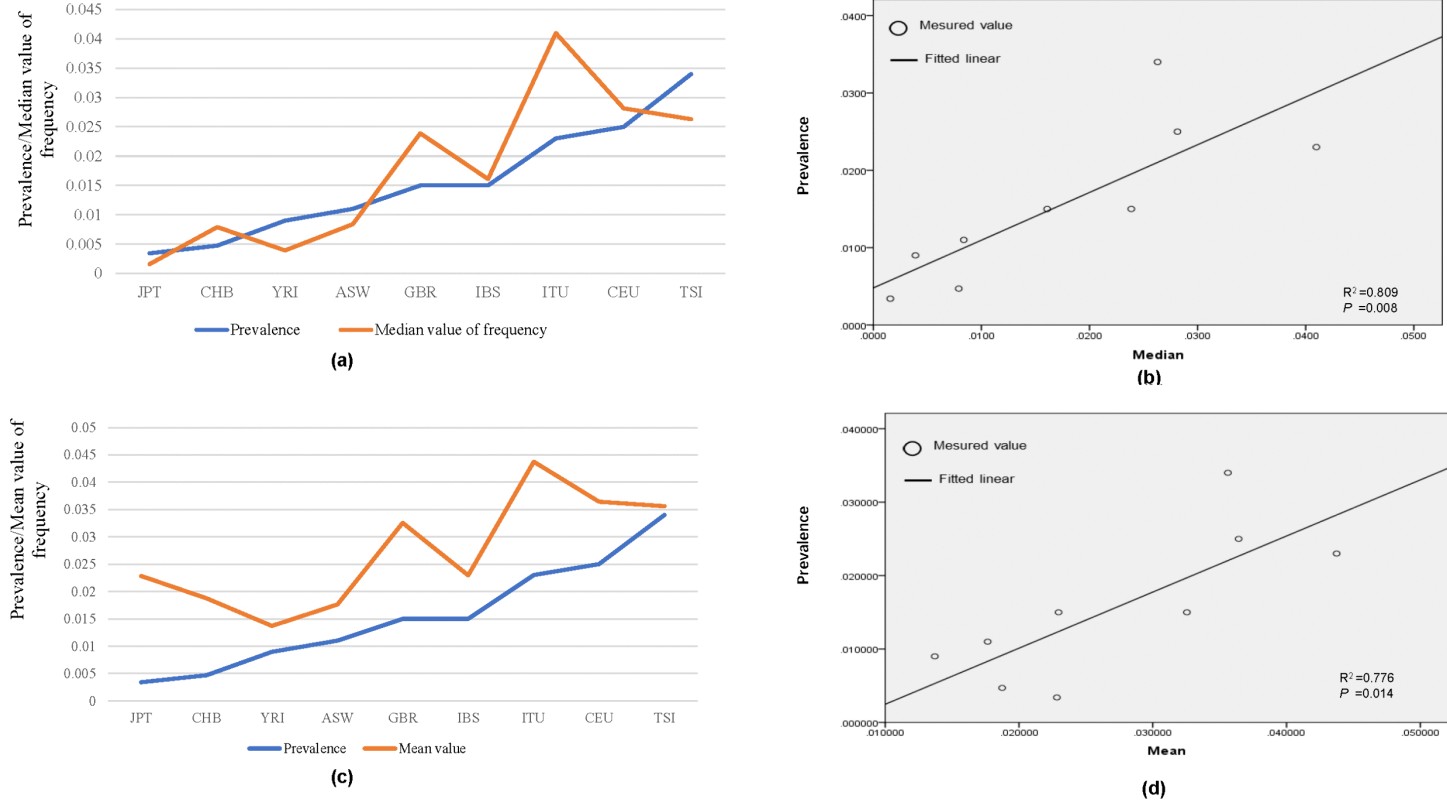

**Fig 2. Genotype frequency and disease prevalence across ethnic groups.** (A) and (C) display the median and mean frequencies of psoriasis-specific genotype combinations and disease prevalence across ethnic groups in line charts, respectively. (B) and (D) show the linear correlation between these median and mean frequencies and disease prevalence, respectively.

method may be specific to psoriasis. These combinations play a crucial role in understanding the genetic architecture of common diseases and can explain the variations in disease prevalence across different ethnicities.

Among the 357 SNPs analyzed (the original association results and the genotype frequencies in the control population for these SNPs were shown in S24,S25 Table), 72 were located in nonsynonymous codons of 41 genes, indicating potential functional consequences on protein structure and function. Additionally, 114 SNPs were found in synonymous codons of 63 genes, suggesting that they may not directly impact protein function but could still affect gene regulation or expression. Furthermore, there were 156 SNPs identified within various functional regions of 81 genes. Notably, an additional set of 15 SNPs was found in the regulatory regions upstream and downstream of specific genes. These regulatory regions play a crucial role in controlling gene expression and could potentially contribute to disease susceptibility (S26 Table).

### The function of 134 genes contained in psoriasis-specific combinations was involved in immune dysregulation and epidermal barrier dysfunction

We performed Gene Ontology (GO) analysis to investigate the molecular functions and biological processes of the 134 genes (S27 Table). The results revealed significant enrichment in antigen presentation, immune-related biological processes and cell-cell adhesion, including "antigen processing and presentation" ($P = 5.13E-6$), "positive regulation of T cell activation" ($P = 3.14E-5$), and "positive regulation of leukocyte cell-cell adhesion" ($P = 6.23E-5$) (Fig 4).

**Table 3. Proportion of the significant linear correlations (P < 0.05) in 1000 permutation tests under different models.**

| Counts of population | Models | Proportion of 100 tests | |
|---|---|---|---|
| | | Proportion (P < 0.05) | Proportion (R² > 0.5) |
| 10 | Complete random model[a] | 52.2% | 39.2% |
| | Constrained model[b] | 64.6% | 54.4% |
| | Constrained lost model[c1] (10%) | 65.4% | 56.2% |
| | Constrained lost model[c2] (20%) | 65.1% | 57.7% |
| | Constrained lost (20%) and mixed (5%) model[d1] | 62.6% | 54.6% |
| | Constrained lost (20%) and mixed (10%) model[d2] | 60.5% | 51.7% |
| | Constrained lost (20%) and mixed (20%) model[d3] | 60.2% | 51.1% |
| | Constrained lost (20%) and mixed (30%) model[d4] | 56.1% | 48.2% |
| | Constrained lost (30%) and mixed (30%) model[d5] | 56.5% | 47.1% |
| 20 | Complete random model[a] | 81.6% | 27.1% |
| | Constrained model[b] | 82.2% | 52.7% |
| | Constrained lost model[c1] (10%) | 79.6% | 51.8% |
| | Constrained lost model[c2] (20%) | 79.6% | 54.2% |
| | Constrained lost (20%) and mixed (5%) model[d1] | 77.7% | 50.7% |
| | Constrained lost (20%) and mixed (10%) model[d2] | 79.6% | 49.5% |
| | Constrained lost (20%) and mixed (20%) model[d3] | 80.4% | 47.0% |
| | Constrained lost (20%) and mixed (30%) model[d4] | 77.4% | 46.2% |
| | Constrained lost (30%) and mixed (30%) model[d5] | 76.2% | 44.5% |
| 50 | Complete random model[a] | 99.7% | 13.8% |
| | Constrained model[b] | 94.9% | 49.4% |
| | Constrained lost model[c1] (10%) | 93.4% | 50.0% |
| | Constrained lost model[c2] (20%) | 92.5% | 48.7% |
| | Constrained lost (20%) and mixed (5%) model[d1] | 94.0% | 47.7% |
| | Constrained lost (20%) and mixed (10%) model[d2] | 93.4% | 51.2% |
| | Constrained lost (20%) and mixed (20%) model[d3] | 91.9% | 44.9% |
| | Constrained lost (20%) and mixed (30%) model[d4] | 92.6% | 46.3% |
| | Constrained lost (30%) and mixed (30%) model[d5] | 91.3% | 43.1% |
| 100 | Complete random model[a] | 100.0% | 6.9% |
| | Constrained model[b] | 98.8% | 49.4% |
| | Constrained lost model[c1] (10%) | 97.9% | 50.1% |
| | Constrained lost model[c2] (20%) | 97.0% | 50.3% |
| | Constrained lost (20%) and mixed (5%) model[d1] | 97.2% | 48.0% |
| | Constrained lost (20%) and mixed (10%) model[d2] | 96.6% | 45.3% |
| | Constrained lost (20%) and mixed (20%) model[d3] | 96.1% | 46.4% |
| | Constrained lost (20%) and mixed (30%) model[d4] | 97.8% | 43.9% |
| | Constrained lost (30%) and mixed (30%) model[d5] | 96.5% | 40.9% |

a. Each permutation test was carried out in 1000 groups of random numbers at the same time; b. Limited random numbers which floating within 20% of the reference value; c1. Limited random numbers which floating within 20% of the reference value, and removed 10% of all the numbers; c2. removed 20% of all the numbers; d1. Limited random numbers which floating within 20% of the reference value, in addition removed 20% and supplemented 5% of all the numbers randomly; d2. removed 20% and supplemented 10%; d3.removed 20% and supplemented 20%; d4.removed 20% and supplemented 30%; d5.removed 30% and supplemented 30%

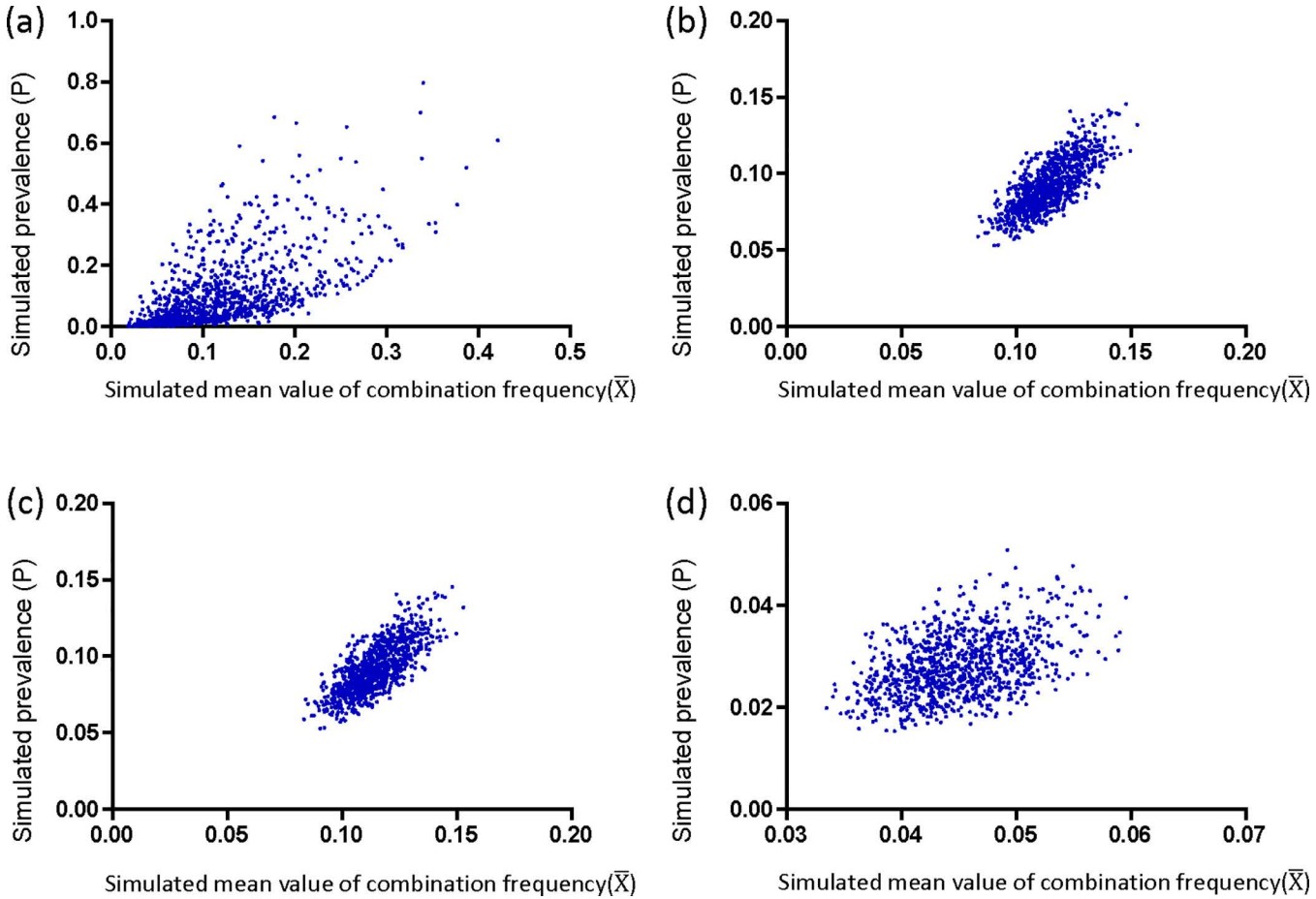

**Fig 3. Genotype combination frequency and disease prevalence across 1000 simulated populations.** Scatter plots illustrating the relationship between mean 3-locus genotype combination frequency and disease prevalence across 1000 simulated populations under different models. The x-axis represents mean genotype combination frequency, and the y-axis represents disease prevalence. (A) Complete random model; (B) constrained model with 20% fluctuation from reference population frequency; (C) constrained lost model with 20% fluctuation and 20% data removal; (D) constrained lost and mixed model with 20% fluctuation, 30% data removal, and 30% data mixing.

## Discussion

The causal genetic architecture of complex traits or diseases is influenced by multiple variants [16]. Previous studies have shown that combining multiple disease-associated SNPs is more effective than single SNP-based analysis in identifying associations between complex diseases and genetic predisposition [17,18]. In this study, we employed whole exome sequencing data and the CGCP program, which utilizes an exhaustive algorithm to search for specific genetic signatures associated with the disease. After eliminating background noise, we identified 620 genotype combinations specific to psoriasis that merged into 303 combinations of variants of genes.

We conducted stable computational simulations to verify the stability of the linear correlation between P and $\overline{X}$. By implementing rigorous quality control measures and data limitations, as well as designing different mixed and lost ratios, we ensured the stability of the simulated linear relationship (Table 3). These computer simulations confirmed that the genotype combinations identified through the CGCP method play a critical role in determining disease prevalence across diverse populations. To further enhance the validity of our findings and minimize potential errors, we introduced a

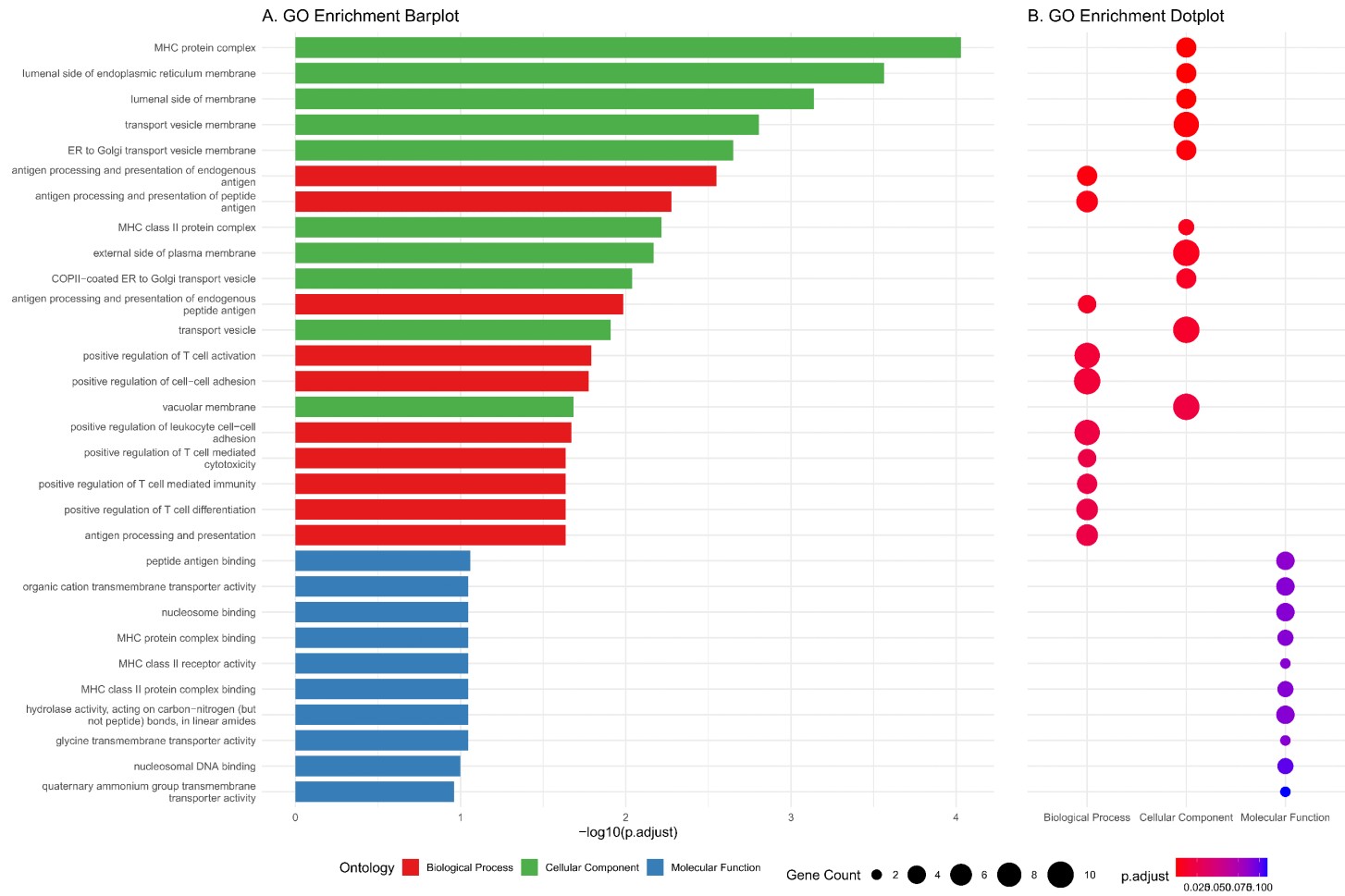

**Fig 4. GO enrichment of 134 genes.** Dotplot showing enriched GO terms of 134 genes divided by genes collected in the pathway.

constrained lost and mixed model. With 10 simulated populations, significant linear correlations were observed in more than 60% of our 100 tests (S4 Fig).

Even with a smaller number of populations, the scatter points generated by P and $\overline{X}$ may align along the major axis of an ellipse, which is consistent with our study findings. We selected 9 populations with significant differences in disease prevalence for our study. We observed a linear relationship between disease prevalence and the genotype frequencies identified by CGCP in each respective population. However, when the frequencies were too close to each other due to random computer simulation, the scatter points generated by P and $\overline{X}$ appeared clustered in a circular pattern. In such cases, the linear relationship might not be statistically significant (S5 Fig).

The use of 3 genotypes as the unit of combination was initially motivated by computational efficiency. However, through mathematical deduction (Supplementary_ mathematical derivation), we discovered that these combinations of 3 genotypes can also simulate a perfect linear relationship with the incidence rate of disease. This finding suggests that these combinations play a crucial role in the pathogenesis of the disease.

Traditionally, genetic disorders have been categorized as either Mendelian diseases or common diseases. Mendelian disorders are typically rare and result from deleterious mutations in a single gene with high penetrance, such as sickle-cell anemia and cystic fibrosis. These deleterious mutations tend to have low population frequencies due to purifying

(negative) selection [19–21]. Under the concept of mutation-selection balance, deleterious variants in the population are maintained through a balance between new mutations and the purging effect of purifying selection, ensuring that Mendelian disorders with high penetrance remain at low prevalence levels [22–24]. Weghorn et al. demonstrated through computer simulations and data-driven approaches that even in the presence of genetic drift, deleterious mutations undergoing purifying selection continue to adhere to the principles of the mutation-selection balance hypothesis [25]. In comparison to Mendelian diseases, common diseases exhibit more complex patterns of inheritance. Apart from rare variants that have noticeable deleterious effects, the genetic architecture of common diseases is likely characterized by a substantial number of common variants with minor to moderate effects [26–30]. Unlike highly deleterious mutations, which experience strong negative selection pressure [31–33], slight mutations with small to modest effects may undergo weaker or even no selection [34,35].

The neutral theory of molecular evolution (NTME), originally proposed by Kimura in 1968 and formalized in 1983, provides a framework for understanding the role of neutral or nearly neutral mutations in molecular evolution [36–38]. According to the framework of NTME, Nei et al. have argued that a significant proportion of morphological evolution is driven by neutral or nearly neutral mutations, which have a major impact on both phenotypic and molecular evolution. Natural selection plays a lesser role in shaping these evolutionary processes [39–42]. Disease-associated variants and their frequencies in populations are influenced by the evolutionary processes of genetic drift and natural selection [43].

In our study, through CGCP analysis of psoriasis, we found that the frequency of all genotypes in psoriasis-specific combinations was relatively common (≥1%). Additionally, the majority of these variants appeared to be neutral or had minimal deleterious effects based on annotation information. Interestingly, we observed a correlation between the increased prevalence of common diseases and the expansion of these disease-specific common variants.

While slightly deleterious variants may have hitchhiking effects with adaptive mutations through positive selection, resulting in a higher probability of retention in the population [44–48], our study suggests that the majority of the variants observed in psoriasis-specific combinations are likely to be selectively neutral or have minimal deleterious effects [49,50]. Furthermore, it is worth noting that the majority of common diseases do not appear to impact an individual's adaptation to the environment. This suggests that disease-specific common variants are more likely influenced by genetic drift rather than negative selection.

Although negative selection affects the genetic architecture of common diseases, the majority of common variants observed in modern populations are more likely to be selectively neutral [43,51]. Therefore, these variants may accumulate in the population through genetic drift, allowing slightly deleterious alleles to increase in frequency. Our findings, in line with the predictions of the neutral theory of molecular evolution (NTME), suggest that genetic drift plays a crucial role in the evolution of common diseases. Moreover, considering the rising prevalence of common diseases, it becomes evident that genetic drift is deeply intertwined with their evolutionary dynamics.

In our study, we identified 303 combinations of variants of genes specific to psoriasis, consisting of 134 genes. Among these genes, 41 have previously been associated with psoriasis and are involved in antigen presentation, immune response regulation and cell-cell adhesion. The functional enrichment of these genes further highlights the central roles of immune dysregulation and epidermal barrier dysfunction in psoriasis. Genes such as HLA-B, HLA-C, HLA-DQA1, and HLA-DRA are established risk factors; their encoded MHC class I/II molecules drive T-cell activation via aberrant autoantigen presentation, thereby initiating inflammatory cascades [52]. Additionally, CD74, an MHC class II chaperone, regulates antigen processing efficiency, with its dysregulation documented in psoriatic lesions [53]. IL12B is a key driver of psoriasis; its encoded IL-12 promotes Th1 differentiation and synergizes with IL-23 to amplify Th17-mediated inflammation, while activation of the NLRP3 inflammasome mediates the secretion of proinflamatory cytokines such as IL-1β [54,55]. Furthermore, genes implicated in lymphocyte activation and macrophage recruitment, such as CD27, CD6, and AIF1, may regulate psoriatic inflammation [56,57]. Regarding intercellular adhesion, the gene CDSN encodes a cornified envelope protein pivotal for epidermal barrier integrity. Meanwhile, EVPL mediates epidermal cell-cell junctions, and DDR1 regulates cell

adhesion and migration. Collectively, dysfunction in these genes may contribute to psoriasis pathogenesis by impairing barrier function [58]. Further investigation of these genes and pathways could enhance our understanding of the underlying mechanisms of psoriasis.

Although the CGCP method is a valuable approach for exploring the genetic architecture of complex diseases and identifying combinations of variants of genes, there are ways to improve its effectiveness. First, increasing the sample size with sequencing data can enhance the ability to filter out false positive combinations and obtain more accurate genotype frequencies within a specific population. Second, conducting larger-scale studies on disease prevalence across different ethnic groups at different time points and sub-types would provide valuable insights. Lastly, this study focused on three-locus combinations, which may be missing and underestimating the higher order patterns. In future work, we aim to optimize computational resources to analyze combinations involving more than three SNPs, enabling a more comprehensive exploration of complex genetic interactions and a clearer understanding of the true causal combinations underlying the disease.

In conclusion, we have successfully utilized the CGCP method to search for genetic signatures of a common disease, specifically psoriasis. Through this approach, we identified 134 genes that are specific to psoriasis and contribute to the variability in prevalence across different populations. This strategy provides a novel perspective on understanding the population genetics characteristics of common diseases.

## Key points

- Firstly, employing a novel strategy known as the CGCP method (Causal Genotype Combination Pattern), previously published, we analyzed a whole-exome sequencing dataset encompassing 781 psoriasis cases and 676 healthy controls from the Chinese Han population. This analysis revealed the identification of 620 genotype combinations specific to psoriasis, exclusively present in affected individuals at frequencies ranging from 4.7% (37) to 10% (78) of all cases. Furthermore, these genotypes collectively aggregated into 134 genes, with 41 of them having previously been associated with psoriasis.

- Secondly, leveraging the publicly available dataset from the 100 Genomes Project Phase III and prevalence data on psoriasis across various ethnic populations obtained through literature review, we established a robust positive correlation and formulated a linear regression model ($y = 61.72x + 0.48$, 95% CI [21.60, 101.84]) between the average frequency of these psoriasis-specific genotype combinations (x) and the disease prevalence in different ethnic groups (y). This model may provide insight into the varying prevalence of psoriasis across diverse populations.

- Thirdly, the frequency of each genotype for all psoriasis-specific combinations that we identified was found to be common (≥ 1%), and most variants appeared to be neutral or have minimal deleterious effects, as inferred from the annotation information.

## Supporting information

**S1 Fig. Frequency distribution of psoriasis-specific genotype combinations.** Distribution of psoriasis-specific genotype combination frequencies across ethnic groups (ASW, CEU, CHB, GBR, IBS, ITU, JPT, TSI, YRI), shown in panels (a) to (i). The x-axis represents genotype combination frequency, and the y-axis represents frequency counts, exhibiting positive skew.
(TIF)

**S2 Fig. Ethnic variations in genotype frequency and disease prevalence.** (a), (c), (e), and (g) present line charts of median genotype combination frequencies and disease prevalence across ethnic groups for different selection criteria. (b),

(d), (f), and (h) depict linear correlations between these frequencies and disease prevalence. Genotypes combinations in (a) and (b) were selected from psoriasis-specific combinations which appeared in less than 30 and more than 20 patients; Genotypes combinations in (c) and (d) were selected from psoriasis-specific combinations which appeared in less than 37 and more than 30 patients; Genotypes combinations in(e) and (f) were made up by the variants which generated from CGCP analysis in 338 "cases" and 338 controls in disease-associated loci; Genotypes combinations in (g) and (h) were made up by the variants which generated from CGCP analysis in 781 cases and 676 controls in disease-uncorrelated loci. (TIF)

**S3 Fig. GO analysis of biological processes and molecular functions.** Annotations derived from Gene Ontology (GO) analysis. (a) Biological process annotations; (b) molecular function annotations. (TIF)

**S4 Fig. Linear relationships in constrained mixed and lost model.** Analysis of genotype combinations with 30% loss and mixing in a constrained model. (A) Significant linear relationships in 10 populations; (B) non-significant linear relationships. (TIF)

**S5 Fig. Scatter patterns in simulated frequency data.** (A) Scatter plot of simulated genotype frequencies (P and $\overline{X}$) forming a concentrated circle when frequencies are closely clustered from random simulation. (B) Elliptical scatter with significant linear relationship when frequencies are dispersed. (TIF)

**S1 Table. Association analysis in 781 cases and 676 controls after quality control in disease-associated loci.** (XLSX)

**S2 Table. CGCP analysis in first filter stage which observed in more than 20 patients with psoriasis.** (XLSX)

**S3 Table. Psoriasis-associated region reported by GWAS Catalog.** (XLSX)

**S4 Table. 101, 657 variants covered by exon-sequencing falls into psoriasis-related regions.** (XLSX)

**S5 Table. Association analysis in 338 "patients" and 338 controls after quality control in disease-associated loci.** (XLSX)

**S6 Table. CGCP analysis in 338 "patients" and 338 controls in disease-associated loci for identifing background noise.** (XLSX)

**S7 Table. Annotated each SNPs in genotype combinations into correspondingly genes in 338 "patients" and 338 controls in disease-associated loci.** (XLSX)

**S8 Table. Association analysis in 781 cases and 676 controls after quality control in disease-uncorrelated loci.** (XLSX)

**S9 Table. CGCP analysis in781 patients and 676 controls in disease-uncorrelated loci for identifing background noise.** (XLSX)

**S10 Table. Annotated each SNPs in genotype combinations into correspondingly genes in 781 patients and 676 controls in disease-uncorrelated loci.**
(XLSX)

**S11 Table. Annotated each SNPs into combinations of variants of genes after removed threshold of background noise of variants combinations.**
(XLSX)

**S12 Table. Psoriasis-specific genotype combinations were merged in 303 combinations of variants of genes.**
(XLSX)

**S13 Table. The frequency of psoriasis-specific genotype combinations in different ethnics.**
(XLSX)

**S14 Table. The frequency of genotype combinations in different ethnics in noise background.**
(XLSX)

**S15 Table. Parameters of linear regression equations established between mean value of combination frequency and prevalence of disease in 1000 permutation tests under the complete random mode.**
(XLSX)

**S16 Table. Parameters of linear regression equations established between mean value of combination frequency and prevalence of disease in 1000 permutation tests under the constrained model.**
(XLSX)

**S17 Table. Parameters of linear regression equations established between combination frequency and prevalence of disease in 1000 permutation tests under the Constrained with lost 10% model.**
(XLSX)

**S18 Table. Parameters of linear regression equations established between combination frequency and prevalence of disease in 1000 permutation tests under the Constrained with lost 20% model.**
(XLSX)

**S19 Table. Parameters of linear regression equations established between combination frequency and prevalence of disease in 1000 permutation tests under the Constrained with lost 20% and mixed 5% model.**
(XLSX)

**S20 Table. Parameters of linear regression equations established between combination frequency and prevalence of disease in 1000 permutation tests under the Constrained with lost 20% and mixed 10% model.**
(XLSX)

**S21 Table. Parameters of linear regression equations established between combination frequency and prevalence of disease in 1000 permutation tests under the Constrained with lost 20% and mixed 20% model.**
(XLSX)

**S22 Table. Parameters of linear regression equations established between combination frequency and prevalence of disease in 1000 permutation tests under the Constrained with lost 20% and mixed 30% model.**
(XLSX)

**S23 Table. Parameters of linear regression equations established between combination frequency and prevalence of disease in 1000 permutation tests under the Constrained with lost 30% and mixed 30% model.**
(XLSX)

**S24 Table. Association analysis in 357 variants within psoriasis-specific combinations.**
(XLSX)

**S25 Table. The frequency of risk genotypes within psoriasis-specific combinations in controls of our whole exome sequencing dataset.**
(XLSX)

**S26 Table. The type of mutaions and locations of 357 variants in psoriasis-specific combinations.**
(XLSX)

**S27 Table. Detailed GO enrichment information for 134 genes.**
(XLSX)

**S28 Table. Source data and statistical results of the 1000-permutation linear correlation test under the Completely Random Model.** The data were generated using the command python completely_random_time.py.txt 1000 20 1. This model assumes independent population generation without constraints. The first sheet contains raw data from the first permutation (from 0.origin.txt and 0.txt), including 15 random numbers, genotype combination frequencies, and the prevalence (P), Mean (X_bar), and Median (X_md) for 20 populations. The subsequent sheets provide the final linear regression analysis results across all 1000 permutations (from result.txt).
(XLSX)

**S29 Table. Source data and statistical results of the 1000-permutation linear correlation test under the Constrained Model.** This model simulates gene flow by limiting the variation of disease-specific genotype frequencies across ethnicities to 20% of the reference population. The data were generated using the command python convergence_loss_times_origion.py.txt 1000 20 20 0 0 1. The first sheet detailed the first permutation data (from 0.origin.txt and 0.txt), and the following sheets present the final linear regression analysis results between genotype frequencies and disease prevalence across 1000 tests (from result.txt), demonstrating the impact of constrained frequency fluctuations.
(XLSX)

**S30 Table. Source data and statistical results of the 1000-permutation linear correlation test under the Constrained and Lost Model.** This model incorporates a filtering mechanism where a specified proportion (e.g., 30%) of the genotype combinations derived from the pathogenic combinations are "lost" or excluded across all populations. The simulation was performed using the command python convergence_loss_times_origion.py.txt 1000 20 20 30 0 1. The first sheet provides details of the first permutation, showing how the random loss of combinations in the initial population is consistently applied to others, Despite the reduction in combination elements, the results in the subsequent sheets demonstrate that the matrix-form linear correlation between genotype frequency and disease prevalence remains robust across the 1000 permutation tests.
(XLSX)

**S31 Table. Source data and statistical results of the 1000-permutation linear correlation test under the Constrained, Lost, and Mixed Model.** This model simulates the most realistic scenario by introducing "fake combinations" (Type-I error noise) that are not derived from pathogenic genotypes but are output by the CGCP method due to limited control group size. The simulation used the command python convergence_loss_times_origion.py.txt 1000 20 20 30 10 1, incorporating a 30% loss rate and a 10% mix rate of fake combinations. The first sheet detailed the first permutation data (from 0.origin.txt and 0.txt), and the subsequent sheets present the final statistical results across 1000 permutations (from result.txt). While the introduction of noise makes this model deviate from the ideal linear formula, it provides a crucial comparison with previous models to evaluate the robustness of the correlation between genotype frequency and disease prevalence in real-world settings.
(XLSX)

**S1 File. Supplementary file computer program simulation.**
(DOCX)

**S2 File. Supplementary file mathematical derivation.**
(DOCX)

## Author contributions

**Data curation:** Weiran Li, Wanrong Wang, Kejia Wu, Feiran Zhou.

**Formal analysis:** Yuanjing Zhang.

**Funding acquisition:** Xiaodong Zheng.

**Investigation:** Yuanjing Zhang, Weiran Li, Wanrong Wang, Kejia Wu, Feiran Zhou.

**Methodology:** Xiaodong Zheng.

**Project administration:** Xiaodong Zheng.

**Resources:** Xiaodong Zheng.

**Supervision:** Xiaodong Zheng.

**Validation:** Yuanjing Zhang.

**Visualization:** Yuanjing Zhang, Wanrong Wang.

**Writing – original draft:** Yuanjing Zhang.

**Writing – review & editing:** Xiaodong Zheng.

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
