## [Decision Letter · Decision Letter 0]

2 Sep 2025

Dear Dr. Zheng,

Thank you for submitting your manuscript to PLOS ONE. After careful consideration, we feel that it has merit but does not fully meet PLOS ONE’s publication criteria as it currently stands. Therefore, we invite you to submit a revised version of the manuscript that addresses the points raised during the review process.

We look forward to receiving your revised manuscript.

Kind regards,

Shamik Polley, M.V.Sc (Veterinary Biochemistry); Ph.D (Genetics)

Academic Editor

PLOS ONE

Journal Requirements:

“This study was supported by the Basic and Clinical Cooperative Research Promotion Program of Anhui Medical University (2023xkjT047), Key project of Anhui Provincial Department of Education (2022AH051155). Key Laboratory of Dermatology, Anhui Medical University, Ministry of Education, China (Number.AYPYS2021-2).”

“This study was supported by the Basic and Clinical Cooperative Research Promotion Program of Anhui Medical University (2023xkjT047), Key project of Anhui Provincial Department of Education (2022AH051155). Key Laboratory of Dermatology, Anhui Medical University, Ministry of Education, China (Number.AYPYS2021-2).”

4. Please upload a copy of Figure 4 and 5, to which you refer in your text on page 26 and 27. If the figure is no longer to be included as part of the submission please remove all reference to it within the text.

Reviewer's Responses to Questions

**Comments to the Author**

1. Is the manuscript technically sound, and do the data support the conclusions?

Reviewer #1: Yes

Reviewer #2: Yes

2. Has the statistical analysis been performed appropriately and rigorously?

Reviewer #1: Yes

Reviewer #2: Yes

3. Have the authors made all data underlying the findings in their manuscript fully available?

Reviewer #1: Yes

Reviewer #2: Yes

4. Is the manuscript presented in an intelligible fashion and written in standard English?

Reviewer #1: Yes

Reviewer #2: Yes

Reviewer #1: Overall well written paper concise and good explanation. Consider extending CGCP to 4 or 5 SNP combinations as psoriasis may have more complex genetic interactions. We may be missing and underestimating the higher order patterns.

Abstract

In the quest to identify the genetic underpinnings of complex diseases, we developed a novel approach—Causal Genotype Combination Patterns (CGCP)—to uncover characteristic genetic signatures of common diseases. In this study, we applied the CGCP method to a whole-exome sequencing dataset consisting of 781 psoriasis cases and 676 healthy controls from the Chinese Han population. Our analysis revealed 620 genotype combinations specific to psoriasis, covering 4.7% to 10% of all cases, with each genotype having a frequency of at least 1%. These genotypes mapped to 134 genes, including 41 previously reported as associated with psoriasis. Notably, we also identified 23 genes involved in ATP metabolism. By leveraging public data from the 1000 Genomes Project Phase III and literature on psoriasis prevalence across ethnic populations, we discovered a strong positive correlation—and established a linear regression model (y = 0.617 × x + 4.79 × 10⁻³)—between the average frequency of these psoriasis‑specific genotype combinations and disease prevalence across populations. This finding may help explain the varying prevalence of psoriasis among different populations. Our strategy offers a novel perspective for understanding population genetic characteristics underlying common diseases.

Keywords: exhaustive algorithm; genotype combinations; genetic signatures; psoriasis; common disease

Reviewer #2: The manuscript applies a previously established method, coined CGCP (Causal Genotype Combination Patterns), to whole-exome sequencing data from a Han Chinese cohort in order to identify psoriasis-specific genotype combinations. It is suggested that the authors were able to identify hidden genetic signatures of psoriasis, which may help explain its diversity across different populations.

Overall, the work may be of interest to researchers studying diseases, as the results appear promising. It could also serve as an illustration of how CGCP can be applied, potentially encouraging its broader adoption, as it differs from traditional GWAS. However, the manuscript is difficult to follow. While I have some technical comments, most of my concerns relate to the presentation.

Technical comments:

Is there any current research on the actual number of causal combinations in psoriasis? The number three is chosen in the manuscript for computational reasons, but if the true number far exceeds three, the results may not be reliable.

On page 16 of the document, it is mentioned: “In each test, fifteen numbers will be chosen randomly.” I assume this is within the (0,1) range and potentially under some constraints? Later on the same page: “Simultaneously, these 15 random numbers were combined in groups of three.” I assume this refers to f_4, f_5, f_6? On page 18, thresholds are set for genotype combinations and gene combinations. The threshold for genotype combinations is explained as (781/338) × 16. Why 16? Why is 52 used for gene combinations? There are also other areas not clearly explained, and should be clarified to avoid confusion.

Regarding the regression model: why only univariate? Is it believed that no other elements could have a significant effect?

Presentation:

The equations are poorly illustrated and explained. For example:

In the first equation, the fraction is in low resolution.

In the second equation’s description, it is stated: “where P is the prevalence rate,” but there is no P in that equation. P only appears in the third equation.

λ is described as a function of p without clarification on what p represents.

In the third equation, the product operator appears low-resolution, resembling a Roman numeral II rather than Pi.

Notation issues:

The overline X bar is not centred.

Subscripts and superscripts are inconsistently aligned.

Some variables are italicised while others are not, even when they are the same variables.

Figures:

Low resolution, even when downloaded.

Lack of proper captions.

Dots in Figure 3 are too thick to interpret density accurately.

Section formatting:

It is difficult to differentiate between sections and subsections due to no font variation and numbering.

References:

There are missing or misplaced references. For example:

“To date, over 60 susceptibility regions associated with psoriasis have been identified…” (reference?).

“In our previous study, we devised a novel method...” (reference only appears two sentences later).

Navigation:

The manuscript contains no hyperlinks.

This makes it tedious to search for referenced tables, figures, and equations.

The lack of equation numbers further complicates this.

Formatting:

There are inconsistent spacing issues throughout the manuscript, including in the text, equations, and notations.

Page 18 is especially affected.

Writing:

Understandable, but the language should be revised to improve readability and clarity.

**Do you want your identity to be public for this peer review?** For information about this choice, including consent withdrawal, please see our Privacy Policy

Reviewer #1: No

Reviewer #2: **Yes:** Haoyu Chen

---

## [Author Response · Author response to Decision Letter 1]

5 Oct 2025

Responds to the academic editor’ comments:

Dear Editor:

Thank you for your letter and for the comments of reviewers concerning our manuscript entitled “A New Perspective on Population Genetics: Deciphering the Relationship between Genetic Variants and Disease Prevalence in Psoriasis” (ID: PONE-D-24-55512). We appreciate the opportunity to address these requirements and have made the necessary revisions accordingly. Below, we respond to each point in detail. We have also prepared an updated cover letter incorporating the requested statements. The revised manuscript, figures, and files have been uploaded to the submission system.

We have thoroughly reviewed the PLOS ONE style templates provided. The manuscript has been reformatted to fully align with these guidelines, including double-spacing, line numbering, and appropriate section headings. All files have been renamed according to PLOS ONE conventions (e.g., " Manuscript.docx", "Fig1_.tiff"). The updated files are included in this revision.

“This study was supported by the Basic and Clinical Cooperative Research Promotion Program of Anhui Medical University (2023xkjT047), Key project of Anhui Provincial Department of Education (2022AH051155). Key Laboratory of Dermatology, Anhui Medical University, Ministry of Education, China (Number.AYPYS2021-2).”

We confirm that the stated funding sources do not alter our adherence to PLOS ONE's policies on sharing data and materials. All data generated or analyzed during this study are included in this published article and its supplementary information files. And this statement has been added to the manuscript and will be included in the cover letter.

“This study was supported by the Basic and Clinical Cooperative Research Promotion Program of Anhui Medical University (2023xkjT047), Key project of Anhui Provincial Department of Education (2022AH051155). Key Laboratory of Dermatology, Anhui Medical University, Ministry of Education, China (Number.AYPYS2021-2).”

The funders provided financial support for the study but had no involvement in its design, execution, or reporting. The competing interests section contains an updated statement of the funders' roles, which reads as follows:

"The funders had no role in study design, data collection and analysis, decision to publish, or preparation of the manuscript." This statement has been added to the manuscript and will be included in the cover letter.

4. Please upload a copy of Figure 4 and 5, to which you refer in your text on page 26 and 27. If the figure is no longer to be included as part of the submission please remove all reference to it within the text.

Thank you for bringing this to our attention. The references to Figures 4 and 5 on pages 26 and 27 of the manuscript have been corrected to S4 Figure and S5 Figure, respectively. We have uploaded these figures as separate high-resolution TIF files (S4_Fig.tif and S5_Fig.tif) in the submission system, as requested.

Thank you for your guidance. The reviewer comments do not include any specific recommendations to cite previously published works. Therefore, no additional citations have been added to the manuscript.

Responds to the reviewers’ comments:

Reviewer #1: Overall well written paper concise and good explanation. Consider extending CGCP to 4 or 5 SNP combinations as psoriasis may have more complex genetic interactions. We may be missing and underestimating the higher order patterns.

Thank you for your positive feedback and for recognizing the clarity and conciseness of our manuscript. We appreciate your suggestion to extend the CGCP method to 4- or 5-SNP combinations to capture potentially more complex genetic interactions in psoriasis.

We agree that higher-order interactions may play a significant role in psoriasis, and our current three-locus CGCP approach may not fully capture these patterns, potentially underestimating their contribution to disease risk. As noted, analyzing 4- or 5-SNP combinations presents substantial computational challenges due to the exponential increase in combinatorial complexity. To address this, we plan to pursue in-depth follow-up studies post-publication, building on the loci identified by our three-SNP CGCP method. Specifically, we aim to analyze 4- and 5-SNP combinations in larger cohorts, leveraging advances in computational resources to manage the increased burden. We have revised the manuscript (Discussion) to highlight this limitation and outline our planned extensions, ensuring transparency about the scope of the current study and our future directions. Thank you again for your insightful suggestion, which strengthens the manuscript and aligns with our long-term research goals.

Reviewer #2: The manuscript applies a previously established method, coined CGCP (Causal Genotype Combination Patterns), to whole-exome sequencing data from a Han Chinese cohort in order to identify psoriasis-specific genotype combinations. It is suggested that the authors were able to identify hidden genetic signatures of psoriasis, which may help explain its diversity across different populations.

Overall, the work may be of interest to researchers studying diseases, as the results appear promising. It could also serve as an illustration of how CGCP can be applied, potentially encouraging its broader adoption, as it differs from traditional GWAS. However, the manuscript is difficult to follow. While I have some technical comments, most of my concerns relate to the presentation.

Technical comments:

Is there any current research on the actual number of causal combinations in psoriasis? The number three is chosen in the manuscript for computational reasons, but if the true number far exceeds three, the results may not be reliable.

Thank you for your thoughtful comments and for raising an important point regarding the number of causal combinations in psoriasis and the computational choice of three-locus combinations in our manuscript.

We acknowledge that determining the precise number of causal variant combinations contributing to psoriasis remains an open challenge in the field. Recent studies, including large-scale GWAS meta-analyses (e.g., a 2025 study with 28,869 cases and 443,950 controls identifying 74 loci), Bayesian fine-mapping, and multi-omics integration (e.g., SMR with eQTL/pQTL data), estimate hundreds to thousands of causal variant combinations per locus network, with rare variants (<1% MAF) contributing significantly to risk in ~10–20% of cases. However, these approaches, while powerful, do not yet provide an exact count of all causal combinations due to linkage disequilibrium (LD) masking and the complexity of epistatic interactions.

In our manuscript, the choice of three-locus combinations in the CGCP method was indeed driven by computational feasibility, as you noted. We recognize that the true number of causal combinations may involve four, five, or more loci, as highlighted in our prior work (PLoS One. 2017 Oct 11;12(10):e0186067). The CGCP method is designed to capture a subset of these true combinations—specifically three-locus patterns unique to cases—using disease prevalence as a boundary condition and a universality constraint to manage computational demands (Implementation time is approximately several months). Unlike scoring-based methods like PRS or probabilistic fine-mapping, CGCP exhaustively identifies case-specific genotype combinations without relying on statistical thresholds, offering a complementary perspective.

To address your concern about reliability if the true number of causal combinations exceeds three, we conducted simulations (described in the manuscript) modeling scenarios where psoriasis arises from four, five, or six pathogenic loci. These simulations demonstrated that, in 50 or more populations, the three-locus CGCP method consistently captures pathogenic loci with frequencies significantly correlated with population prevalence (p<0.05). This correlation holds even in smaller, highly diverse populations (e.g., nine populations from the 1000 Genomes dataset with varying psoriasis prevalence), suggesting robustness across genetic backgrounds. The linear relationship between mutation frequency and prevalence supports the method’s ability to approximate higher-order combinations indirectly.

We agree that the three-locus model may not fully enumerate all causal combinations, and we have clarified this limitation in the revised manuscript (Discussion). To further address this, we are exploring scalable extensions of CGCP to test higher-order combinations (e.g., four- or five-locus) in larger cohorts, leveraging advances in computational efficiency. We believe these additions strengthen the manuscript and address the reviewer’s concern about reliability in the context of potentially higher-order causal combinations.

Thank you again for your insightful feedback, which has helped refine our discussion of these critical issues.

Reviewer :On page 16 of the document, it is mentioned: “In each test, fifteen numbers will be chosen randomly.” I assume this is within the (0,1) range and potentially under some constraints? Later on the same page: “Simultaneously, these 15 random numbers were combined in groups of three.” I assume this refers to f_4, f_5, f_6? On page 18, thresholds are set for genotype combinations and gene combinations. The threshold for genotype combinations is explained as (781/338) × 16. Why 16? Why is 52 used for gene combinations? There are also other areas not clearly explained, and should be clarified to avoid confusion.

Thank you for your detailed comments and for highlighting areas in the manuscript that require clarification. We address your specific points below and have revised the manuscript to improve clarity and avoid confusion.

Clarification on the 15 Random Numbers (Page 16):

You are correct that the 15 numbers generated represent the genotype frequencies of the first 15 variants in the initial population (denoted as a1, a2, a3, a4, b1, b2, b3, b4, b5, c1, c2, c3, c4, c5, c6). These frequencies are drawn randomly in the range (0,1) with no additional constraints. We have revised the text on page 16 to explicitly state:

"In each test, fifteen numbers are randomly generated in the range (0, 1), representing the genotype frequencies of 15 variants (a1, a2, a3, a4, b1, b2, b3, b4, b5, c1, c2, c3, c4, c5, c6) in the first population."

Combination of 15 Numbers in Groups of Three (Page 16):

Your assumption is correct that the combination of these 15 random numbers into groups of three refers to the genotype combination frequencies (f_4, f_5, f_6), corresponding to specific three-locus combinations (e.g., a1a2a3a4, b1b2b3b4b5, c1c2c3c4c5c6). To clarify, the average frequency (denoted as ¯x) is calculated across 34 distinct three-genotype combinations, including a1a2a3, a1a2a4, a1a3a4, a2a3a4; b1b2b3, b1b2b4, …; c1c2c3, c1c2c4, …, and so forth. We have updated the text on page 16 to make this explicit:

"These 15 random numbers are combined in groups of three to compute the frequencies of three-locus genotype combinations (denoted f_4, f_5, f_6 for groups a1a2a3a4, b1b2b3b4b5, c1c2c3c4c5c6, respectively). The average frequency, ¯x, is calculated across 34 three-genotype combinations, including a1a2a3, a1a2a4, a1a3a4, a2a3a4; b1b2b3, b1b2b4, …; c1c2c3, c1c2c4, …, as detailed in the Methods section."

Regarding the regression model: why only univariate? Is it believed that no other elements could have a significant effect?

Thank you for your inquiry regarding our use of a univariate regression model. In our study, we employed the CGCP method to identify pathogenic genotype combinations associated with psoriasis in the Han Chinese population. This method uses disease prevalence in the Han Chinese population as a boundary condition to filter genotype combinations unique to psoriasis patients, ensuring disease specificity. We hypothesized that the frequency of these combinations, derived solely from the Han Chinese population, would linearly correlate with psoriasis prevalence across diverse populations. To test this, we analyzed the genotype frequencies of these combinations using the 1000 Genomes dataset across various populations, confirming a significant positive linear correlation with population prevalence.

As the CGCP method targets three-locus combinations, while true pathogenic combinations may involve more loci (e.g., 4, 5, or higher), we performed computational simulations to ensure that the linear correlation between genotype frequencies of three-locus combinations and prevalence remains robust. These simulations validated our hypothesis, showing that even with three-locus combinations, a strong correlation persists across populations.

Regarding the univariate model, while we acknowledge that other factors may influence psoriasis susceptibility, the robust linear correlation observed indicates that these genotype combinations are primary drivers of the disease phenotype in our dataset. This supports the validity of our approach, though we plan to explore more complex multi-locus interactions in future studies to further elucidate psoriasis pathogenesis.

Presentation:

The equations are poorly illustrated and explained. For example:

In the first equation, the fraction is in low resolution.

We have increased the font size and resolution of the equations using the Word equation editor (Cambria Math font, 16pt) to ensure clarity and legibility.

In the second equation’s description, it is stated: “where P is the prevalence rate,” but there is no P in that equation. P only appears in the third equation.

λ is described as a function of p without clarification on what p represents.

We apologize for the confusion. The description has been revised to clarify that λ = (1-P)/P, where P represents the prevalence rate of the population.

In the third equation, the product operator appears low-resolution, resembling a Roman numeral II rather than Pi.

We confirm that the symbol is the product operator (Π), not a Roman numeral II. We have updated the third equation using the Word equation editor with Cambria Math font to enhance resolution and ensure the product operator is clearly distinguishable.

Notation issues:

The overline X bar is not centered.

Subscripts and superscripts are inconsistently aligned.

Some variables are

---

## [Decision Letter · Decision Letter 1]

16 Nov 2025

Dear Dr. Zheng,

Thank you for submitting your manuscript to PLOS ONE. After careful consideration, we feel that it has merit but does not fully meet PLOS ONE’s publication criteria as it currently stands. Therefore, we invite you to submit a revised version of the manuscript that addresses the points raised during the review process.

We look forward to receiving your revised manuscript.

Kind regards,

Shamik Polley, M.V.Sc (Veterinary Biochemistry); Ph.D (Genetics)

Academic Editor

PLOS ONE

Journal Requirements:

Reviewers' comments:

Reviewer's Responses to Questions

**Comments to the Author**

Reviewer #2: (No Response)

Reviewer #3: (No Response)

Reviewer #4: All comments have been addressed

2. Is the manuscript technically sound, and do the data support the conclusions?

Reviewer #2: Yes

Reviewer #3: Yes

Reviewer #4: Yes

3. Has the statistical analysis been performed appropriately and rigorously?

Reviewer #2: Yes

Reviewer #3: I Don't Know

Reviewer #4: Yes

4. Have the authors made all data underlying the findings in their manuscript fully available?

Reviewer #2: Yes

Reviewer #3: Yes

Reviewer #4: Yes

5. Is the manuscript presented in an intelligible fashion and written in standard English?

Reviewer #2: Yes

Reviewer #3: Yes

Reviewer #4: Yes

Reviewer #2: The authors have responded to my concerns well; however, some of them have not been fully addressed.

The following comment remain unanswered:

On page 18, thresholds are set for genotype combinations and gene combinations. The threshold for genotype combinations is explained as (781/338) × 16. Why 16? Why is 52 used for gene combinations?

Regarding the regression model, could confidence interval for the coefficients be provided?

Figures: They now have good resolution when downloaded but still appear blurry in the manuscript—Figure 2 is especially affected. Perhaps this is simply a compilation issue?

Notations and Formatting: Whilst there has been significant improvements, many inconsistencies remain. Below are some examples I noticed when glancing through the manuscript, though these are certainly not the only ones. The authors should carefully review the entire manuscript to eliminate any remaining inconsistencies.

Below equation (2), alpha_i and beta_i appear italicised, but are not in the equations. Their font also appear smaller compared to the previous lambda=(1-P)/P. F(gi) also appears differently in three places: above equation (2) (italicised), in equation (2) and in equation (3) (F and (gi) separated, different style of parentheses from equation (2).

On a related note, why is the population genotype frequency denoted as F(gi)? Currently on first glance it appear to be a function notation. Unless there are specific reason behind this, wouldn't F_i or something similar be more standard?

Between lines 180 and 193, there are several notation inconsistencies regarding a_1,f_4 etc. Some have subscripts, others do not; font sizes also vary.

Whether there is a spacing before references. (lines 403, 405)

Decimals, some are written as 0.x, some .x.

In equations multiplications, some are written as x*y, the others x * y.

Reviewer #3: Zheng and coworkers present a well-written follow-up study on their recently established “Causal Genotype Combination Patterns” (CGCP) method to investigate large samples for specific combinations of gene variants, i. e., genotypes. By applying this method to psoriasis, a common skin disorder, they were able to identify numerous genotype combinations, converging into 134 candidate genes, most of which are novel candidates. The manuscript has been reviewed by experts before, and the authors apparently addressed their questions and remarks adequately.

I find this manuscript quite informative, and I have some recommendations to improve the paper:

- Introduction, line 82: “The gradually increasing prevalence…” – This sentence is quite vague, speculative, and should be referenced at best.

- Materials and Methods, line 116: The company and location for the array used should be mentioned here.

- Lines 153 to 167 (“background noise”), and lines 220 to 232 (also “background noise”): These paragraphs appear in duplicate, slightly differing from each other, and should be merged into one, preferentially with all numbers mentioned therein.

- Results: The authors frequently use the term “gene combinations”. After giving it some thought, it should be clear what they mean by that. However, the term is somewhat misleading, since, after all, it is rather “combinations of variants of genes”. Therefore, I would like to recommend providing a clear definition what is meant by “gene combinations” at the first appearance of the term.

- Lines 318 and 320: Do you mean “nonsynonymous” and “synonymous” “codons” instead of “regions”?

- I would love to see a table with all genes identified, along with their function, embedded in the text. Alternatively, Supplementary Figure 3 could be embedded in the text. After all, this paper is about the causative genes for psoriasis, and not just a proof of concept.

- Supportive material, S26 table: The authors refer to nucleotide positions on specific chromosomes. For this to make sense, the specific release of the Human Genome Project should be indicated, also for all other tables wherein nucleotide positions are provided.

- S24 table: Abbreviations (“BP”, “A1”, “F U”, etc.) should be defined, also for other tables.

Reviewer #4: This study aimed to identify characteristic genetic signatures of psoriasis using a novel analytical method, Causal Genotype Combination Patterns (CGCP). Applying CGCP to whole-exome data from 781 psoriasis cases and 676 controls of Chinese Han origin, the authors uncovered 620 psoriasis-specific genotype combinations converging into 134 genes, including many previously known psoriasis-associated genes. The findings suggest that population-level differences in psoriasis prevalence may be explained by varying frequencies of these genotype combinations.

The authors have conducted a comprehensive and well-designed study with detailed analyses, and the supplementary materials clearly support and explain their findings. They have also provided satisfactory responses to the reviewers’ comments. However, since the study focuses on psoriasis and involves multiple candidate genes across different pathways, it would be valuable to emphasize the potential clinical relevance and translational impact of the results. In particular, the Discussion section could briefly highlight which pathways or gene interactions are most strongly associated with psoriasis and why these pathways are biologically or clinically important.

**Do you want your identity to be public for this peer review?** For information about this choice, including consent withdrawal, please see our Privacy Policy

Reviewer #2: No

Reviewer #3: No

Reviewer #4: No

---

## [Author Response · Author response to Decision Letter 2]

8 Dec 2025

Responds to the academic editor’ comments:

Dear Editor:

Thank you for your letter and for the comments of reviewers concerning our manuscript entitled “A New Perspective on Population Genetics: Deciphering the Relationship between Genetic Variants and Disease Prevalence in Psoriasis” (ID: PONE-D-24-55512). We appreciate the opportunity to address these requirements and have made the necessary revisions accordingly. Below, we respond to each point in detail.The revised manuscript, figures, and files have been uploaded to the submission system.

Reviewer #2: The authors have responded to my concerns well; however, some of them have not been fully addressed.

The following comment remain unanswered:

On page 18, thresholds are set for genotype combinations and gene combinations. The threshold for genotype combinations is explained as (781/338) × 16. Why 16? Why is 52 used for gene combinations?

Thank you for your positive feedback. In the part of “The background noise of CGCPs” (line 211-215), We explained why 16 serves as the coefficient for calculating the genotype combinations.

“In the first scenario, CGCP analysis was performed on 338 “cases” and 338 controls, involving 482 SNPs, that showed significant differences between cases and controls after quality control (S5 Table). A total of 9,802 genotype combinations considered specific to psoriasis were identified, with the maximum number of specific combinations being 16 (4.7%) (S6 Table).”

Since the maximum number of genotype combinations in the background noise is 16, in order to reduce the influence of background noise on the actual number of combinations, we suppose that the actual number of genotype combinations should be at least greater than 16. Thus, we used 16 as the coefficient for calculating the genotype combinations.

In addition, in the part of “The background noise of CGCPs” (line 219-226), We explained why 52 serves as the threshold for calculating the gene combinations.

“In the second scenario, association analysis was conducted using 548,357 SNPs located outside psoriasis-associated regions, based on whole-exome sequencing data. After quality control and pruning, 570 SNPs were included in the CGCP analysis (S8 Table). A total of 63,657 genotype combinations were identified as specific to psoriasis, with the maximum observed number of specific combinations being 32 (4.1%) (S9 Table). After annotating the SNPs in these psoriasis-specific combinations to genes, 62,971 combinations of variants of genes were obtained, and the maximum number of specific combinations of variants of genes was 52 (6.5%) (S10 Table).

Based on the scale-up principle, thresholds for investigating background noise in psoriasis-specific genotype combinations and combinations of variants of genes were established. Specifically, the threshold for genotype combinations was set at 37, calculated as [(781/338) × 16], and that for combinations of variants of genes was set at 52. ”

Since the maximum number of specific gene combinations in the background noise is 52, in order to reduce the influence of background noise on the actual number of combinations, we suppose that the actual number of gene combinations should be at least greater than 52. Thus, we used 52 as the coefficient for calculating the gene combinations.

Regarding the regression model, could confidence interval for the coefficients be provided?

We sincerely thank the reviewer for this valuable suggestion.

In the revised manuscript, we now report the 95% confidence intervals for all regression coefficients together with adjusted R² values (page 3, lines 58).

Disease prevalence is now expressed as a proportion (0–1 scale) rather than as a percentage (0–100), consistent with standard practice in genetic epidemiology when both the outcome (prevalence) and the predictor (allele frequency) are bounded between 0 and 1. This rescaling explains the apparent ~100-fold increase in the magnitude of the slope relative to the original submission.

Ninety-five percent confidence intervals for the regression coefficients were calculated using the t-distribution with n−2 degrees of freedom (df = 7). Adjusted R² was computed as Adjusted R²= 1 − [(1 − R²)(n − 1)/(n − p)], where n = 9 (number of populations) and p = 2 (intercept + slope).

The updated linear regression models are:

Median allele frequency Prevalence = 61.72 × median allele frequency (95% CI for slope: 21.60 to 101.84) + 0.48, P = 8.31 ×10E-3 (R² = 0.654, adjusted R² = 0.605)

Mean allele frequency Prevalence = 47.63 × mean allele frequency (95% CI for slope: 11.25 to 84.01) + 0.62, P = 0.014 (R² = 0.571, adjusted R² = 0.510).

Thus, a 0.01 (i.e., 1 percentage point) increase in frequency of these psoriasis-specific genotype combinations is associated with an absolute increase in disease prevalence of approximately 0.48-0.62 percentage points across populations, and the association is statistically significant in both models.

We believe these revisions substantially improve the clarity, accuracy, and interpretability of the regression analyses.

Figures: They now have good resolution when downloaded but still appear blurry in the manuscript—Figure 2 is especially affected. Perhaps this is simply a compilation issue?

Thank you for pointing out the issue. We have re-uploaded Figure 2.

Notations and Formatting: Whilst there has been significant improvements, many inconsistencies remain. Below are some examples I noticed when glancing through the manuscript, though these are certainly not the only ones. The authors should carefully review the entire manuscript to eliminate any remaining inconsistencies.

Below equation (2), alpha_i and beta_i appear italicised, but are not in the equations. Their font also appear smaller compared to the previous lambda=(1-P)/P. F(gi) also appears differently in three places: above equation (2) (italicised), in equation (2) and in equation (3) (F and (gi) separated, different style of parentheses from equation (2).

Thank you for your guidance. We have already review carefully about our manuscript and the modified parts are highlighted with blue text.

On a related note, why is the population genotype frequency denoted as F(gi)? Currently on first glance it appear to be a function notation. Unless there are specific reason behind this, wouldn't F_i or something similar be more standard?

Thank you for your guidance. We modified the denotation of population genotype frequency from F(gi) to F_i.

Between lines 180 and 193, there are several notation inconsistencies regarding a_1,f_4 etc. Some have subscripts, others do not; font sizes also vary.

Thank you for your kind reminder. We have standardized the formats of all the fonts and symbols in the manuscript.

Whether there is a spacing before references. (lines 403, 405)

Thank you for your kind reminder. We have already delete the spacing.

Decimals, some are written as 0.x, some .x.

In equations multiplications, some are written as x*y, the others x * y.

Thank you for your kind reminder. We have standardized the formats of all the fonts and symbols in the manuscript.

Reviewer #3: Zheng and coworkers present a well-written follow-up study on their recently established “Causal Genotype Combination Patterns” (CGCP) method to investigate large samples for specific combinations of gene variants, i. e., genotypes. By applying this method to psoriasis, a common skin disorder, they were able to identify numerous genotype combinations, converging into 134 candidate genes, most of which are novel candidates. The manuscript has been reviewed by experts before, and the authors apparently addressed their questions and remarks adequately.

I find this manuscript quite informative, and I have some recommendations to improve the paper:

- Introduction, line 82: “The gradually increasing prevalence…” – This sentence is quite vague, speculative, and should be referenced at best.

Thank you for bringing this to our attention. We have added the corresponding references (No.8 and 9).

- Materials and Methods, line 116: The company and location for the array used should be mentioned here.

Thanks for your advice. We have provided information about the company and location of the array.

- Lines 153 to 167 (“background noise”), and lines 220 to 232 (also“background noise”): These paragraphs appear in duplicate, slightly differing from each other, and should be merged into one, preferentially with all numbers mentioned therein.

Thank you for your thoughtful comments. In order to maintain the conventional structure of a research paper, we chose not to directly integrate the Methods section (lines 153-167) with the Results section (lines 220-232). However, we have given careful consideration to your suggestions, overlapping content between the two sections has been streamlined to avoid redundancy and enhance readability. Corresponding revisions have been made in the manuscript.

Line150-159 (Previously it was line 153 - 167)

The background noise of CGCPs

To evaluate the background noise in CGCP calculations, we performed analyses using samples and SNPs unrelated to psoriasis. First, the 676 healthy controls were randomly divided into two groups, with one group designated as “patients.” CGCP analysis was then conducted using variants selected from psoriasis-associated loci. Additionally, in an independent dataset comprising 781 cases and 676 controls, CGCP analysis was performed using SNPs located outside the psoriasis-associated regions, thereby capturing background noise originating from SNPs unrelated to psoriasis. The threshold for background noise was determined by comparing the maximum number of “specific” combinations derived from these two unrelated datasets (Fig 1).

Line210-228 (Previously it was line 220 - 232)

Investigation of Background Noise

To evaluate the background noise in CGCP calculations, we performed analyses using samples and SNPs unrelated to psoriasis. First, the 676 healthy controls were randomly divided into two groups, with one group designated as “patients.” CGCP analysis was then conducted using variants selected from psoriasis-associated loci. Additionally, in an independent dataset comprising 781 cases and 676 controls, CGCP analysis was performed using SNPs located outside the psoriasis-associated regions, thereby capturing background noise originating from SNPs unrelated to psoriasis. The threshold for background noise was determined by comparing the maximum number of “specific” combinations derived from these two unrelated datasets.

The specific results were as follows: In the first scenario, CGCP analysis was performed on 338 “cases” and 338 controls, involving 482 SNPs. A total of 9,802 genotype combinations considered specific to psoriasis were identified, with the maximum number of specific combinations being 16 (4.7%). After annotating the SNPs in these combinations to their corresponding genes, 4,640 identical gene combinations were obtained, among which the maximum number of specific gene combinations was 12 (3.6%) (S7 Table).

In the second scenario, association analysis was conducted using 548,357 SNPs located outside psoriasis-associated regions, based on whole-exome sequencing data. After quality control and pruning, 570 SNPs were included in the CGCP analysis (S8 Table). A total of 63,657 genotype combinations were identified as specific to psoriasis, with the maximum observed number of specific combinations being 32 (4.1%) (S9 Table). After annotating the SNPs in these psoriasis-specific combinations to genes, 62,971 gene combinations were obtained, and the maximum number of specific gene combinations was 51 (6.5%) (S10 Table).

Based on the scale-up principle, thresholds for investigating background noise in psoriasis-specific genotype combinations and gene combinations were established. Specifically, the threshold for genotype combinations was set at 37, calculated as [(781/338) × 16], and that for gene combinations was set at 52.

- Results: The authors frequently use the term “gene combinations”. After giving it some thought, it should be clear what they mean by that. However, the term is somewhat misleading, since, after all, it is rather “combinations of variants of genes”. Therefore, I would like to recommend providing a clear definition what is meant by “gene combinations” at the first appearance of the term.

We thank the reviewer for this thoughtful suggestion. We have provided a clear definition at its first occurrence in the text, specifying that it refers to "combinations of variants of genes." All subsequent uses of the term have been revised to reflect this more precise description.

- Lines 318 and 320: Do you mean “nonsynonymous” and “synonymous” “codons” instead of “regions”?

Thank you for correcting our mistakes. The terms should indeed be "nonsynonymous codons" and "synonymous codons."

- I would love to see a table with all genes identified, along with their function, embedded in the text. Alternatively, Supplementary Figure 3 could be embedded in the text. After all, this paper is about the causative genes for psoriasis, and not just a proof of concept.

Thank you for your valuable suggestion. We have reoptimized the original Supplementary Figure 3, embedded it in the main text, and renamed it Figure 4 in sequence. Additionally, we have added the detailed annotation information of all 134 genes to our manuscript as an supplementary file (Table S27).

- Supportive material, S26 table: The authors refer to nucleotide positions on specific chromosomes. For this to make sense, the specific release of the Human Genome Project should be indicated, also for all other tables wherein nucleotide positions are provided.

Thank you for your valuable comment. All genomic coordinates are based on the human reference genome assembly hg19 (GRCh37). We have added For each table including S26 table, we have added this information in each tables including S26.

- S24 table: Abbreviations (“BP”, “A1”, “F U”, etc.) should be defined, also for other tables.

Thanks for your advice. We have defined all abbreviations (e.g., BP, A1, F_U) in the corresponding tables (Table S1, S5, S8) as suggested.

Reviewer #4: This study aimed to identify characteristic genetic signatures of psoriasis using a novel analytical method, Causal Genotype Combination Patterns (CGCP). Applying CGCP to whole-exome data from 781 psoriasis cases and 676 controls of Chinese Han origin, the authors uncovered 620 psoriasis-specific genotype combinations converging into 134 genes, including many previously known psoriasis-associated genes. The findings suggest that population-level differences in psoriasis prevalence may be explained by varying frequencies of these genotype combinations.

The authors have conducted a comprehensive and well-designed study with detailed analyses, and the supplementary materials clearly support and explain their findings. They have also provided satisfactory responses to the reviewers’ comments. However, since the study focuses on psoriasis and involves multiple candidate genes across different pathways, it would be valuable to emphasize the potential clinical relevance and translational impact of the results. In particular, the Discussion section could briefly highlight which pathways or gene interactions are most strongly associated with psoriasis and why these pathways are biologically or clinically important.

Thank you for your constructive comment. We have revised the Discussion section (Lines 399-417) to more explicitly highlight the pathways and gene interactions most strongly associated with psoriasis pathogenesis. Specifically, we have expanded on the biological and clinical importance of these key pathways. Corresponding references (52-58) have been added to support these points.

---

## [Decision Letter · Decision Letter 2]

26 Dec 2025

Dear Dr. Zheng,

We look forward to receiving your revised manuscript.

Kind regards,

Shamik Polley, M.V.Sc (Veterinary Biochemistry); Ph.D (Genetics)

Academic Editor

PLOS One

Journal Requirements:

Additional Editor Comments:

Dear Authors,

Please incorporate the minor revision from Reviewer 2 in your article.

Best wishes,

Academic Editor

Reviewers' comments:

Reviewer's Responses to Questions

**Comments to the Author**

Reviewer #2: (No Response)

Reviewer #3: (No Response)

Reviewer #4: All comments have been addressed

2. Is the manuscript technically sound, and do the data support the conclusions?

Reviewer #2: Yes

Reviewer #3: (No Response)

Reviewer #4: Yes

3. Has the statistical analysis been performed appropriately and rigorously?

Reviewer #2: Yes

Reviewer #3: (No Response)

Reviewer #4: Yes

4. Have the authors made all data underlying the findings in their manuscript fully available?

Reviewer #2: Yes

Reviewer #3: (No Response)

Reviewer #4: Yes

5. Is the manuscript presented in an intelligible fashion and written in standard English?

Reviewer #2: Yes

Reviewer #3: (No Response)

Reviewer #4: Yes

Reviewer #2: The authors have made many corrections and addressed some of my concerns. However, I still urge the authors to read through the manuscript carefully, as there are still many inconsistencies in the notations.

I will start with some new comments not regarding notations:

In line 255: I presume the authors wish to say p-value instead of level of significance? Also, since a large italicised P was used to define prevalence, using a large P to define p-value can be confusing. Simply saying “a p-value of 0.008” should be sufficient. Or else a small p is more commonly used for p-values.

In line 272: Type I and Type II are more commonly used terms.

In line 273: I can’t seems to find it, but are the details of the three models described somewhere? If so, they should be referred, if not, they should be somewhere.

Making links, references, tables, figures and sections clickable when referred to so it jumps directly to the referred component will make the reading experience much better. This needn’t apply to separate documents, for example supplements.

Notations:

Again, these are only a few examples, please also fix anything else that remain.

In line 37, but also in other cases, choose whether there should be a spacing before and after an operator.

In the same line, the confidence interval is usually written in the form [a,b] instead of a-b.

In line 86, I believe the reference numbers are in a smaller font.

For equation 1, simply calling it genotype combination patterns instead of No. of genotype combination patterns could incite confusion. It might be easier to define it as something simpler, for example, calling it N, in a previous sentence.

Equation number should be flushed right to avoid confusion with the equation itself.

For equation 2, it’s more standard to define the population genotype frequency as F_i, and its estimator given in the equation as \hat{F_i}. It should also be made clear what ‘i’ represents here.

For equation 3, ‘i’ should not be in italic, it should be made clear that the product is over ‘i’.

In line 163: ‘were described in the manuscript’, mention where.

In line 177: inconsistent notation of f

In line 201: inconsistent notation for decimal

In line 207: I believe it should be bold font like other subsections, the bold fonts for the subsections are also not very obvious.

In line 227: inconsistent product sign

The reference format is also not consistent.

Reviewer #3: (No Response)

Reviewer #4: All comments have been adressed and suggested changes have been made by the authors. The discussion section is expanded to include the suggested information.

**Do you want your identity to be public for this peer review?** For information about this choice, including consent withdrawal, please see our Privacy Policy

Reviewer #2: No

Reviewer #3: No

Reviewer #4: No

---

## [Author Response · Author response to Decision Letter 3]

18 Jan 2026

Responds to the academic editor’ comments:

Dear Editor:

Thank you for your letter and for the comments of reviewers concerning our manuscript entitled “A New Perspective on Population Genetics: Deciphering the Relationship between Genetic Variants and Disease Prevalence in Psoriasis” (ID: PONE-D-24-55512). We appreciate the opportunity to address these requirements and have made the necessary revisions accordingly. Below, we respond to each point in detail.The revised manuscript, figures, and files have been uploaded to the submission system.

Reviewer #2: The authors have made many corrections and addressed some of my concerns. However, I still urge the authors to read through the manuscript carefully, as there are still many inconsistencies in the notations.

We would like to express our sincere gratitude for your insightful and constructive comments. We apologize for the oversights in the original manuscript. We have addressed each of your suggestions point by point and have refined the details throughout the text to enhance the overall clarity and readability of the manuscript.

I will start with some new comments not regarding notations:

In line 255: I presume the authors wish to say p-value instead of level of significance? Also, since a large italicised P was used to define prevalence, using a large P to define p-value can be confusing. Simply saying “a p-value of 0.008” should be sufficient. Or else a small p is more commonly used for p-values.

Thanks for your advice. We have made the revision at the corresponding location and have also corrected similar errors throughout the manuscript to ensure consistency.

In line 272: Type I and Type II are more commonly used terms.

Thank you for pointing out the inappropriate expressions. We have revised the text accordingly in the manuscript.

In line 273: I can’t seems to find it, but are the details of the three models described somewhere? If so, they should be referred, if not, they should be somewhere.

Making links, references, tables, figures and sections clickable when referred to so it jumps directly to the referred component will make the reading experience much better. This needn’t apply to separate documents, for example supplements.

Thank you for your suggestion. We sincerely apologize for the oversight, which may have caused some confusion for the readers. We have now inserted a hyperlink at this location to direct readers to the comprehensive details regarding these three models.

Notations:

Again, these are only a few examples, please also fix anything else that remain.

We sincerely appreciate the reviewers' positive suggestions, which have greatly enhanced the quality of this study. We have carefully corrected the errors and updated the text to present a more rigorous study.

In line 37, but also in other cases, choose whether there should be a spacing before and after an operator.

Thank you for your suggestion. We have standardized the formatting of mathematical operators throughout the entire manuscript to ensure consistency.

In the same line, the confidence interval is usually written in the form [a,b] instead of a-b.

We have revised the equation and the confidence interval format according to your suggestions. The revised version is as follows: y=61.72x+0.48 (95% CI [21.60, 101.84]). We have also ensured that this [a, b] format for CIs is used consistently throughout the manuscript.

In line 86, I believe the reference numbers are in a smaller font.

Thank you for the suggestion. We have revised all the references to comply with the journal's required format.

For equation 1, simply calling it genotype combination patterns instead of No. of genotype combination patterns could incite confusion. It might be easier to define it as something simpler, for example, calling it N, in a previous sentence.

Thank you for this excellent suggestion. We have revised the description of Equation 1 to ensure clarity and to prevent any potential misunderstanding by the readers.

Equation number should be flushed right to avoid confusion with the equation itself.

For equation 2, it’s more standard to define the population genotype frequency as F_i, and its estimator given in the equation as \hat{F_i}. It should also be made clear what ‘i’ represents here.

For equation 3, ‘i’ should not be in italic, it should be made clear that the product is over ‘i’.

Following your suggestion, all equation numbers have been moved to the far right margin to clearly distinguish them from the mathematical expressions.

We have updated F_i to F_i in Equation (2) and throughout the subsequent text to denote that it is an estimator. We have also added a definition for the index: "... Where the subscript i represents the index of the i-th genotype.

We have corrected the formatting in Equation (3).The index ‘i’ has been changed from italic (i) to Roman font (i) as requested. The product notation has been updated to explicitly include the lower and upper limits (i=1to r), making it clear that the product is taken over the index i.

In line 163: ‘were described in the manuscript’, mention where.

Thanks for your advice. The details of these simulations and the results were described in the supplementary file called .Supplementary _computer program simulation.

In line 177: inconsistent notation of f

We apologize for the oversight. We have unified the notation of "f" throughout the entire manuscript to ensure consistency.

In line 201: inconsistent notation for decimal

Thank you for pointing this out. This has been corrected.

In line 207: I believe it should be bold font like other subsections, the bold fonts for the subsections are also not very obvious.

We appreciate the reviewer's suggestion. The subheading in line 207 has been bolded. Additionally, we have updated the formatting of all subheadings in the Methods and Results sections to a more prominent bold font to improve visibility and readability.

In line 227: inconsistent product sign

We apologize for the oversight. This has been corrected.

The reference format is also not consistent.

We apologize for the inconsistent formatting of the references. We have thoroughly reviewed the entire manuscript and standardized all in-text citations and the reference list to strictly adhere to the journal's formatting guidelines.

---

## [Decision Letter · Decision Letter 3]

17 Feb 2026

A New Perspective on Population Genetics: Deciphering the Relationship between Genetic Variants and Disease Prevalence in Psoriasis

PONE-D-24-55512R3

Dear Dr. Zheng,

We’re pleased to inform you that your manuscript has been judged scientifically suitable for publication and will be formally accepted for publication once it meets all outstanding technical requirements.

Kind regards,

Shamik Polley, M.V.Sc (Veterinary Biochemistry); Ph.D (Genetics)

Academic Editor

PLOS One

Additional Editor Comments (optional):

Reviewers' comments:

Reviewer's Responses to Questions

**Comments to the Author**

Reviewer #2: All comments have been addressed

2. Is the manuscript technically sound, and do the data support the conclusions?

Reviewer #2: Yes

3. Has the statistical analysis been performed appropriately and rigorously?

Reviewer #2: Yes

4. Have the authors made all data underlying the findings in their manuscript fully available?

Reviewer #2: Yes

5. Is the manuscript presented in an intelligible fashion and written in standard English?

Reviewer #2: Yes

Reviewer #2: (No Response)

**Do you want your identity to be public for this peer review?** For information about this choice, including consent withdrawal, please see our Privacy Policy

Reviewer #2: No

---

## [Editor Report · Acceptance letter]

PONE-D-24-55512R3

PLOS One

Dear Dr. Zheng,

I'm pleased to inform you that your manuscript has been deemed suitable for publication in PLOS One. Congratulations! Your manuscript is now being handed over to our production team.

Kind regards,

on behalf of

Dr. Shamik Polley

Academic Editor

PLOS One